# DRAFT-BASED APPROXIMATE INFERENCE FOR LLMS

**Kevin Galim**[1*]   **Ethan Ewer**[2*]   **Wonjun Kang**[1,3]   **Minjae Lee**[1]
**Hyung Il Koo**[1,4]   **Kangwook Lee**[2,5,6]

[1]FuriosaAI   [2]UW-Madison   [3]Seoul National University
[4]Ajou University   [5]KRAFTON   [6]Ludo Robotics

{kevin.galim, kangwj1995, minjae.lee, hikoo}@furiosa.ai
{eewer, kangwook.lee}@wisc.edu

Code: https://github.com/furiosa-ai/draft-based-approx-llm

## ABSTRACT

Optimizing inference for long-context large language models (LLMs) is increasingly important due to the quadratic compute and linear memory cost of Transformers. Existing approximate inference methods, including key-value (KV) cache dropping, sparse attention, and prompt compression, typically rely on coarse predictions of token or KV pair importance. We unify and extend recent work by introducing a framework for approximate LLM inference that leverages small draft models to more accurately predict token and KV pair importance. We provide novel theoretical and empirical analyses justifying lookahead-based importance estimation techniques. Within this framework, we present: (i) **SpecKV**, the first method to use lookahead with a small draft model to enable precise KV cache dropping; (ii) **SpecPC**, which leverages draft model attention activations to identify and discard less important prompt tokens; and (iii) **SpecKV-PC**, a cascaded compression strategy combining both techniques. Extensive experiments on long-context benchmarks demonstrate that our methods consistently achieve higher accuracy than existing baselines while retaining the same efficiency gains in memory usage, latency, and throughput.

## 1   INTRODUCTION

The demand for longer context lengths in large language models (LLMs) (Achiam et al., 2023; Google DeepMind, 2025) continues to grow (Liu et al., 2024b), driven by applications such as dialogue systems (Achiam et al., 2023; Google DeepMind, 2025), document summarization (Liu et al., 2020), and code completion (Du et al., 2024). Modern models like GPT-4 (Achiam et al., 2023) and Gemini 2.5 Pro (Google DeepMind, 2025) have pushed context windows to over a million tokens. However, scaling Transformers (Vaswani et al., 2017) to these lengths remains difficult due to significant computational and memory constraints. Attention computation scales quadratically with context length, increasing inference latency, while key-value (KV) cache memory grows linearly, straining GPU resources. For example, caching the KV states for 128K tokens in Llama-3.1-8B (Grattafiori et al., 2024) can consume over 16GB of memory, limiting the practical scalability of LLMs.

To address scalability challenges, recent work introduces approximate LLM inference techniques that reduce latency and memory usage at inference time. Techniques include sparse attention for prefilling (Jiang et al., 2024a) and decoding (Tang et al., 2024), which speed up inference by having each query attend to only a subset of keys. Sparse prefilling shortens time to the first token, while sparse decoding boosts generation throughput. KV cache dropping (Cai et al., 2024b; Li et al., 2024; Xiao et al., 2024; Zhang et al., 2023) reduces memory and increases throughput by shrinking the cache after prefilling or during decoding. Prompt compression (Choi et al., 2024; Jiang et al., 2023b; Liskavets et al., 2025) further improves efficiency by removing less important tokens before inputting the prompt, reducing both attention and MLP computation, as well as decreasing KV cache size.

---

*Equal contribution.

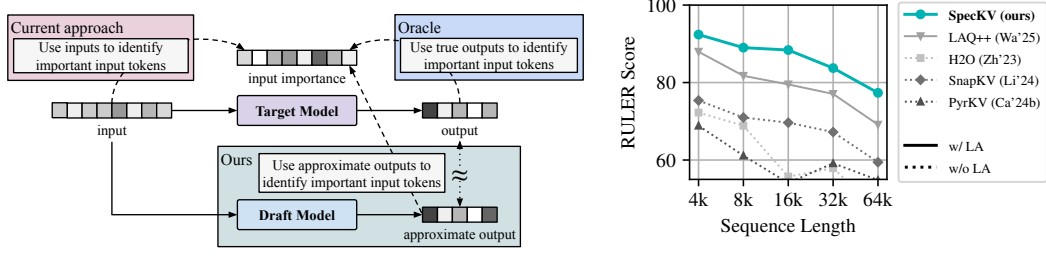

(a) Proposed Framework.  (b) Impact of lookahead (LA) on accuracy.

Figure 1: **(a)** Overview of our Draft-based Approximate Inference framework for input token importance estimation in comparison with current approaches and the oracle approach. Prior methods use input tokens to estimate input token importance. Our approach incorporates draft model predictions of future output tokens, yielding more accurate importance estimates. This better aligns with the hypothetical oracle setting, where the true output is known and influential tokens can be precisely identified. **(b)** On RULER with Llama-3-70B (Grattafiori et al., 2024), lookahead-based methods (LAQ++ (Wang et al., 2025), SpecKV) significantly outperform non-lookahead approaches (H2O (Zhang et al., 2023), SnapKV (Li et al., 2024), PyramidKV (Cai et al., 2024b)), with our proposed SpecKV achieving the best overall downstream score.

Orthogonally, speculative decoding (Cai et al., 2024a; Chen et al., 2023; Hu et al., 2025; Leviathan et al., 2023) accelerates LLM inference by using a small draft model to propose a sequence of multiple tokens, which the target model verifies in parallel. This improves throughput without altering the output distribution and is particularly effective for autoregressive models, where sequential generation is a bottleneck. However, unlike approximate inference, speculative decoding does not lower the total memory or computation requirements and struggles with increasing context length.

In contrast, approximate LLM inference improves efficiency by reducing the amount of computation the model performs. This is often done by estimating the importance of each token or KV pair for future generation and discarding less important ones from attention or feedforward computations. Existing methods (Cai et al., 2024b; Feng et al., 2025; Jiang et al., 2024a; Li et al., 2024) use attention activations from input tokens to predict which tokens or KV pairs future tokens will attend to, as future tokens are not yet available. However, input attention activations alone do not reliably identify the tokens or KV pairs most relevant for future token generation. Recent methods (Liu et al., 2025; Wang et al., 2025) address this by leveraging an approximate output to improve importance estimates. In this work, we unify and extend these techniques by introducing Draft-based Approximate Inference, a lookahead-based framework that uses an inexpensive draft model to approximate future outputs with minimal overhead (Fig. 1a). Our main contributions are as follows:

1. We present Draft-based Approximate Inference, a framework using draft model lookahead for enhanced approximate inference.

2. We present theoretical and empirical analyses justifying lookahead-based KV cache dropping and the use of draft model token importance to approximate target model token importance.

3. Within the Draft-based Approximate Inference framework, we develop two concrete algorithms targeting three LLM inference optimizations: Speculative KV Dropping (SpecKV) for KV cache dropping with sparse prefill, and Speculative Prompt Compression (SpecPC) for prompt compression. Notably, SpecKV is the first to use draft model lookahead for KV cache optimization. Additionally, we introduce SpecKV-PC, a cascaded pipeline integrating both algorithms to achieve superior accuracy, latency, and memory efficiency.

4. We perform comprehensive experiments on long-context benchmarks, demonstrating that our methods attain state-of-the-art accuracy under fixed KV cache or prompt size constraints. Our results consistently outperform prior baselines, underscoring the potential of draft models for fast and accurate approximate inference in large language models.

Table 1: Summary of prior work. Complexity reported without auxiliary/draft model. $n_{in}$ and $n_{out}$ denote the number of input and output tokens, respectively. $C_{max}$ represents the maximum KV cache or prompt capacity. $s_{prefill}$ and $s_{decode}$ indicate the number of keys each query attends to during the prefill and decoding phases, respectively. $L$ is the number of attention layers. Since all time complexities are linear with respect to $L$, we only include $L$ for space complexity. Highlighted cells indicate improved complexity.

| Type | Method | Sparse Attn | KV Dropping | Prefill Time | Decoding Time | Prefill Space | Decoding Space |
|---|---|---|---|---|---|---|---|
| Dense | Dense | ✗ | ✗ | $O(n_{in}^2)$ | $O(n_{out}(n_{in}+n_{out}))$ | $O(Ln_{in})$ | $O(L(n_{in}+n_{out}))$ |
| Sparse attention | MInference, FlexPrefill | Prefill | ✗ | $O(n_{in}s_{prefill})$ | $O(n_{out}(n_{in}+n_{out}))$ | $O(Ln_{in})$ | $O(L(n_{in}+n_{out}))$ |
| | Quest, RetrievalAttention | Decode | ✗ | $O(n_{in}^2)$ | $O(n_{out}s_{decode})$ | $O(Ln_{in})$ | $O(L(n_{in}+n_{out}))$ |
| KV dropping | StreamingLLM | Prefill | Decode | $O(n_{in}C_{max})$ | $O(n_{out}C_{max})$ | $O(\max(n_{in},LC_{max}))$ | $O(LC_{max})$ |
| | H2O | ✗ | Decode | $O(n_{in}^2)$ | $O(n_{out}C_{max})$ | $O(\max(n_{in},LC_{max}))$ | $O(LC_{max})$ |
| | SnapKV, PyramidKV, AdaKV | ✗ | After prefill | $O(n_{in}^2)$ | $O(n_{out}(C_{max}+n_{out}))$ | $O(\max(n_{in},LC_{max}))$ | $O(L(C_{max}+n_{out}))$ |
| | LAQ++ | ✗ | After prefill | $O(n_{in}^2)$ | $O(n_{out}(C_{max}+n_{out}))$ | $O(Ln_{in})$ | $O(L(C_{max}+n_{out}))$ |
| | **SpecKV** (Ours) | Prefill | After prefill | $O(n_{in}s_{prefill})$ | $O(n_{out}(C_{max}+n_{out}))$ | $O(\max(n_{in},LC_{max}))$ | $O(L(C_{max}+n_{out}))$ |
| Prompt compression | LLMLingua-2, CPC, R2C, SpecPrefill, **SpecPC** (Ours) | – | – | $O(C_{max}^2)$ | $O(n_{out}(C_{max}+n_{out}))$ | $O(LC_{max})$ | $O(L(C_{max}+n_{out}))$ |

## 2 RELATED WORK

**Sparse Attention** One way to improve inference efficiency is through sparse attention with static patterns. For example, sliding window attention (Beltagy et al., 2020), used in models like Mistral 7B (Jiang et al., 2023a), Gemma 3 (Gemma Team, 2025), GPT-3 (Brown et al., 2020), and gpt-oss (Agarwal et al., 2025), restricts each query to attend only a fixed-size window of recent keys, reducing computation and KV cache size during decoding. StreamingLLM (Xiao et al., 2024) improves on sliding window by using initial tokens, called attention sinks, along with the sliding window. MInference (Jiang et al., 2024a), adopted by Qwen2.5-1M (Yang et al., 2025b), further boosts prefill efficiency by searching offline for adaptive sparse attention patterns (A-shape, Vertical-Slash, and Block-Sparse) assigned per head. FlexPrefill (Lai et al., 2025) extends this idea by determining sparsity rates for each input prompt. In contrast, Quest (Tang et al., 2024) and RetrievalAttention (Liu et al., 2024a) target the decoding stage by only retrieving the most important KV pairs from the cache, reducing both memory bandwidth and computational demands during generation.

**KV Cache Dropping** KV dropping reduces computation and memory during decoding. Sliding window attention (Beltagy et al., 2020) and StreamingLLM (Xiao et al., 2024) are examples of KV dropping methods (as well as sparse attention) as they permanently evict KV pairs from cache. H2O (Zhang et al., 2023) improves on this by dynamically selecting attention sinks, termed heavy-hitters, using attention scores at each decoding step, while also maintaining a sliding window. SnapKV (Li et al., 2024) compresses the KV cache at the end of the prefill stage by dropping unimportant KV pairs. Subsequent work extends this idea by allocating KV cache budgets non-uniformly across layers (PyramidKV (Cai et al., 2024b)) and attention heads (AdaKV (Feng et al., 2025), HeadKV (Fu et al., 2025)). However, these approaches drop tokens based only on current information, making them less robust to changes in token importance over time (Nawrot et al., 2025). Recently, Wang et al. (2025) proposed Lookahead Q-Cache (LAQ++), which addresses this by generating draft queries with a sparse approximation of the target model, using them to compute more accurate importance scores, though this comes at the cost of no reduction in peak memory usage.

**Prompt Compression** Prompt compression removes tokens before reaching the model, reducing compute and memory usage during both prefill and decoding, unlike KV dropping, which speeds up only decoding. It also surpasses sparse attention by saving both attention and MLP computation. Prompt compression works seamlessly with all types of inference setups, such as APIs or inference engines like vLLM (Kwon et al., 2023), since it does not require any modifications to the model. However, KV dropping can achieve higher compression because it selects tokens per head, while prompt compression drops the same tokens across all layers and heads.

In a question-answer setup, prompt compression may be question-agnostic (compressing context without considering the question) or question-aware (factoring in the question). Selective context (Li et al., 2023) and LLMLingua (Jiang et al., 2023b) are training-free, question-agnostic approaches using a small LLM to keep only key tokens. LongLLMLingua (Jiang et al., 2024b) adapts this

for longer contexts in a question-aware manner. LLMLingua-2 (Pan et al., 2024) trains a small model (Conneau et al., 2020) to score token importance without using the question. CPC (Liskavets et al., 2025) uses a trained encoder to compute sentence importance via cosine similarity with the question, while R2C (Choi et al., 2024) splits the prompt into chunks, processes each with the question using a fine-tuned encoder-decoder Transformer (FiD (Izacard & Grave, 2021)), and ranks them via cross-attention. Similar to our proposed SpecPC, SpecPrefill (Liu et al., 2025) leverages attention scores from a smaller draft model to identify important tokens, using the draft model for lookahead to improve token importance estimates.

**Speculative Decoding** Speculative decoding (Cai et al., 2024a; Chen et al., 2023; Hu et al., 2025; Leviathan et al., 2023) accelerates LLM inference by using a small draft model to propose multiple tokens that the target model verifies in parallel. This increases decoding throughput without changing the output distribution, addressing the bottleneck of autoregressive generation. Previous work further accelerates speculative decoding by enabling approximate inference in the draft model, using techniques such as sparse attention (Sadhukhan et al., 2025), KV cache dropping (Sun et al., 2024), or KV cache quantization (Tiwari et al., 2025), all while preserving exact inference. In contrast, our approach leverages draft models to enable fast, approximate inference directly in the target model.

Table 1 summarizes prior work, highlighting their prefill, decoding time, and memory complexities.

## 3 PROPOSED FRAMEWORK: DRAFT-BASED APPROXIMATE INFERENCE

Previous LLM approximation methods (Cai et al., 2024b; Feng et al., 2025; Jiang et al., 2024a; Li et al., 2024) estimate the importance of current tokens on future generation by analyzing current attention patterns. While this can be effective, it provides only a rough estimate of each token's importance. In contrast, if future tokens were available, we could make substantially better importance estimates by directly identifying which input tokens contribute to generating those output tokens. However, these future tokens are inaccessible before generation.

Recent work has explored using approximate future information to improve token importance estimation. LAQ++ (Wang et al., 2025) extends SnapKV (Li et al., 2024) by generating draft queries and then using them to compute more accurate importance scores. SpecPrefill (Liu et al., 2025) generates lookahead tokens with a small draft model and relies on the draft model's attention to those tokens to estimate input token importance. We bring these ideas together under a unified framework, Draft-based Approximate Inference that leverages approximate future information to improve token importance estimation. We further extend this framework with two new algorithms: SpecKV for KV cache dropping and sparse prefilling (Section 4.1) and SpecPC for prompt compression (Section 4.2). Finally, we introduce cascaded compression with SpecKV-PC, combining both approaches into a single pipeline for superior accuracy and efficiency (Section 4.3).

While our methods use draft models, a technique also common in speculative decoding, our objective is fundamentally different. Speculative decoding improves hardware utilization by having a draft model propose tokens that the target model verifies, accelerating generation without changing the output distribution. However, it does not reduce total computation or memory usage. In contrast, our framework reduces computation and memory costs by approximating the target model.

### 3.1 JUSTIFICATION FOR LOOKAHEAD-BASED KV CACHE DROPPING

KV cache dropping requires estimating the importance of each input KV pair. We define importance as the average attention activation from output queries to each input key. Specifically, the vector of importance scores and its approximation are given by

$$s^T = \frac{1}{n_{\text{out}}} \sum_{i=1}^{n_{\text{out}}} \text{Softmax}\left(\frac{x_i^{(o)T} W_q W_k^T X^T}{\sqrt{d}}\right) \quad \text{and} \quad \hat{s}^T = \frac{1}{n_{\text{out}}} \sum_{i=1}^{n_{\text{out}}} \text{Softmax}\left(\frac{\hat{x}_i^{(o)T} W_q W_k^T X^T}{\sqrt{d}}\right),$$

(1)

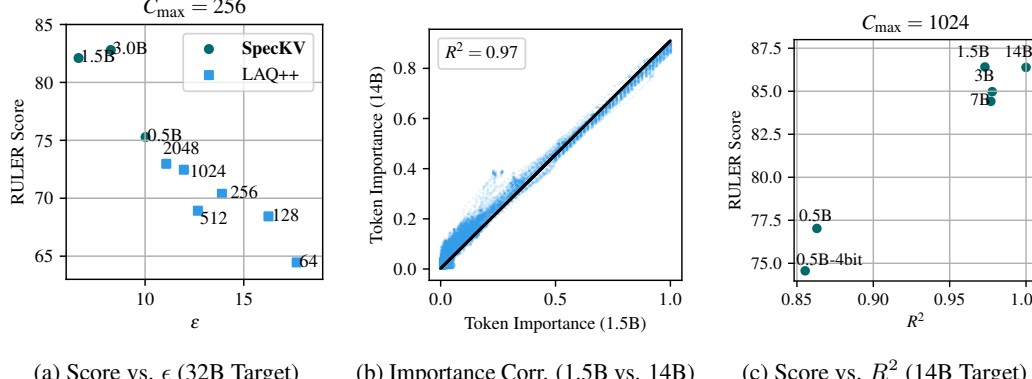

(a) Score vs. $\epsilon$ (32B Target)  (b) Importance Corr. (1.5B vs. 14B)  (c) Score vs. $R^2$ (14B Target)

Figure 2: Experimental validation on RULER-32K tasks (5 samples each) using Qwen2.5 models. **(a)** Lower error $\epsilon$ (Eq. (2)) yields higher downstream scores. Increasing the draft model size (SpecKV) or initial cache size (LAQ++)[1] reduces $\epsilon$, with SpecKV outperforming LAQ++. **(b)** Importance scores (as used in SpecPC) of the draft and target models are highly correlated. **(c)** For SpecPC, a larger draft model improves both the token importance correlation ($R^2$) and the final task performance.

where $X = [x_1, \ldots, x_{n_{in}}]^T \in \mathbb{R}^{n_{in} \times d}$ is the matrix of input embeddings, $x_i^{(o)} \in \mathbb{R}^d$ is the input embedding from the $i$th output token, and $\hat{x}_i^{(o)} \in \mathbb{R}^d$ is its approximation. $s_i$ and $\hat{s}_i$ denote the importance of the $i$th KV pair.

To understand when lookahead-based KV cache dropping algorithms provide reliable importance estimates, we analyze how error in approximate input embeddings impacts error in importance score estimates for a single attention head.

**Theorem 1.** *If* $\|x_i^{(o)} - \hat{x}_i^{(o)}\|_2 \leq \epsilon$ *for all $i$ and* $\|x_j\|_2 \leq \sqrt{d}$ *for all $j$, then* $\|s - \hat{s}\|_2 \leq \epsilon \|W_q W_k^T\|_2$.

This result shows that for a single attention layer, the worst-case error in the approximate importance scores is proportional to the worst-case error in the approximate input embeddings, implying that lookahead-based KV cache dropping algorithms provide reliable importance estimates as long as the draft model remains reasonably accurate (see Section B.1 for proof).

To evaluate the quality of different draft outputs, we compute $\epsilon$ for two algorithms (LAQ++ and SpecKV). While Theorem 1 assumes a strict token-wise alignment (where each draft token $\hat{x}_i^{(o)}$ corresponds to a target token $x_i^{(o)}$), in practice, the draft and target models often generate sequences of different lengths ($\hat{n_{out}} \neq n_{out}$). To address this mismatch, we relax the strict point-wise condition and empirically evaluate draft quality using the global distance between the centroids of the output sequences. We compute $\epsilon$ as:

$$\epsilon = \left\| \frac{1}{n_{out}} \sum_{i=1}^{n_{out}} x_i^{(o)} - \frac{1}{\hat{n_{out}}} \sum_{i=1}^{\hat{n_{out}}} \hat{x}_i^{(o)} \right\|_2. \tag{2}$$

This metric serves as a practical proxy for the distributional mismatch between the target and draft outputs. We report $\epsilon$ averaged across all attention layers. As shown in Fig. 2a, increasing draft quality (decreasing $\epsilon$) leads to higher downstream accuracy. Furthermore, Section E.4 demonstrates how SpecKV's lookahead mechanism improves importance score correlation and accuracy, particularly for long outputs.

### 3.2 JUSTIFICATION FOR DRAFT-BASED PROMPT COMPRESSION

Most KV cache dropping algorithms estimate token importance using attention scores from the target model itself. While effective, this approach is too computationally expensive for tasks like prompt compression, where we need to estimate token importance without forwarding the entire

---

[1]For LAQ++, initial cache size refers to the value of $C_{max}$ used during the sparse draft generation.

input through the target model. A more efficient alternative is to use a smaller draft model for this estimation. To justify this approach, we study how the similarity between the draft and target models' outputs correlates with the similarity of their attention activations in a single attention layer.

The target model attention layer uses weights $W_q$, $W_k$, and $W_v$, and the draft model attention layer uses $\hat{W}_q$, $\hat{W}_k$, and $\hat{W}_v$. Let the input prompt be $X = [x_1, \ldots, x_n]^T \in \mathbb{R}^{n \times d}$. The outputs of the target attention layer and its approximation are

$$Y = \text{Softmax}\left(\frac{XW_qW_k^TX^T}{\sqrt{d}}\right)XW_v = AXW_v, \quad \hat{Y} = \text{Softmax}\left(\frac{X\hat{W}_q\hat{W}_k^TX^T}{\sqrt{d}}\right)X\hat{W}_v = \hat{A}X\hat{W}_v, \tag{3}$$

where $X = [x_1, \ldots, x_n]^T \in \mathbb{R}^{n \times d}$, $A = [a_1, \ldots, a_n]^T$ is the attention matrix, and $\hat{A} = [\hat{a}_1, \ldots, \hat{a}_n]^T$ is the approximate attention matrix.

If the scaled inputs satisfy the Restricted Isometry Property (RIP)[2] (Candes & Tao, 2005), a condition widely studied in compressed sensing to ensure the stable recovery of sparse signals, we can establish the following bound:

**Theorem 2.** *If there exists a constant $c$ such that $cX^T$ satisfies the Restricted Isometry Property with parameters $(2k, \delta)$, where $\delta$ is the restricted isometry constant and $k$ is the approximate sparsity of $a_i$ and $\hat{a}_i$, and the output error satisfies $\|y_i - \hat{y}_i\|_2 \leq \epsilon\|X\|_{\infty,2}$, then the attention error satisfies $\|a_i - \hat{a}_i\|_2 \leq \frac{2c\epsilon\|X\|_{\infty,2}}{\sigma_{\min}(W_v)(1-\delta)}$.[3]*

This result offers a surprising and elegant connection: it reveals that mathematical tools developed for compressed sensing can also bound the error in attention approximations. Specifically, it shows that the worst-case error in the approximate attention activations is proportional to the worst-case error in the approximate outputs, with the constant depending on the conditioning of the weight matrices and the maximum input embedding norm. This implies that if the draft model provides a reasonable approximation of the output, it also gives a reasonable approximation of the attention activations (see Section B.2 for proof). Furthermore, even if the scaled inputs do not satisfy the RIP, we can still bound the attention approximation error by applying Theorem 3 (see Section B.3 for proof).

In addition to the theoretical analysis, we examine the correlation between the importance scores (as computed by SpecPC) for Qwen2.5-Instruct (1.5B as draft, 14B as target). In Fig. 2b, we plot the draft importance scores against the corresponding target importance scores. The results reveal a strong correlation, supporting the use of draft attention activations to approximate target token importance. Furthermore, this correlation strengthens as the draft model size increases (Fig. 2c).

## 4 DRAFT-BASED APPROXIMATE INFERENCE METHODS

### 4.1 SPECKV: ROBUST IMPORTANCE ESTIMATION FOR KV CACHE DROPPING

Existing sparse attention and KV cache dropping methods, such as SnapKV, estimate token importance by analyzing recent attention activations. This approach can be inaccurate when the set of important KV pairs shifts during generation, as past patterns do not always predict future ones. We argue that a more robust estimate can be derived from the attention activations of draft queries for future tokens.

LAQ++ attempts this by generating draft queries from the target model using an initially compressed cache. These queries are then used to compute more accurate importance scores for a second, more informed compression pass. However, this two-pass method provides no reduction in peak memory because it must store the entire original KV cache of the target model to avoid recomputing the expensive prefill stage.

To overcome this limitation, we propose SpecKV. Our method employs a lightweight draft model to generate the draft output, which substantially reduces the cost of the lookahead step. This enables an accurate compression of the KV cache without sacrificing peak memory reduction.

---

[2]The input embedding matrix may satisfy the RIP if its entries are approximately uniformly or normally distributed. RIP can also hold with positional embeddings constructed from a Fourier basis.

[3]$\|X\|_{\infty,2}$ denotes the maximum $\ell_2$ norm of $X$'s rows; $\sigma_{\min}(W_v)$ is the smallest singular value of $W_v$.

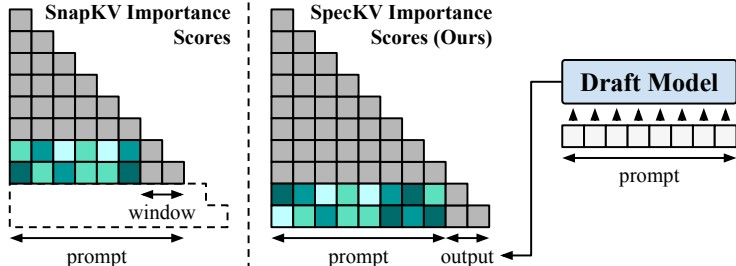

Figure 3: Overview of SpecKV: Instead of using only the last prompt tokens like SnapKV, SpecKV employs a lightweight draft model to generate lookahead tokens, providing richer context for more accurate KV importance estimation. Tokens in window are always retained.

SpecKV (Algorithm 1) begins by generating a draft output of length $n_{\text{lookahead}}$ using a small draft model, which acts as a proxy for the target model's future outputs. During prefilling, both the input tokens and the draft tokens are passed through the target model. For each attention head, we compute token importance scores by measuring the cross-attention between the queries from the last $n_{\text{window}}$ input tokens and the draft output tokens to the remaining input keys (Fig. 3). We apply local pooling with kernel size $k$ to the attention scores to maintain continuity. These scores guide two optimizations: sparse prefilling and KV cache dropping. For sparse prefilling, we use a variation of the Vertical-Slash kernel pattern introduced in (Jiang et al., 2024a). For KV cache dropping, we retain the top $C_{\text{max}} - n_{\text{window}}$ KV pairs with the highest importance scores, along with the final $n_{\text{window}}$ KV pairs from the most recent tokens.

## 4.2 SpecPC: Leveraging Draft Models for Efficient Prompt Compression

SpecKV leverages the draft model outputs to enable more effective KV cache dropping. However, greater efficiency gains are possible by leveraging more information from the draft model. As demonstrated in Section 3.2, draft model attention scores serve as reliable estimates of target token importance. Building on this insight, we introduce SpecPC, an extension of SpecPrefill (Liu et al., 2025), which compresses the prompt to reduce latency and memory usage during both prefilling and decoding, surpassing the efficiency benefits of traditional KV cache dropping.

SpecPC (Algorithm 2) feeds an input prompt (length $n_{\text{in}}$) to the draft model and directly extracts its attention activations $A \in \mathbb{R}^{n_{\text{layer}} \times n_{\text{head}} \times (n_{\text{in}} + n_{\text{lookahead}} - 1) \times n_{\text{in}}}$, where $n_{\text{layer}}$ and $n_{\text{head}}$ denote the number of layers and heads. These activations indicate token importance and are used to drop less relevant tokens from the prompt.

SpecPrefill (Liu et al., 2025) uses a window size of $n_{\text{window}} = 1$, meaning it relies on attention scores from queries associated with the last input token and $n_{\text{lookahead}}$ draft output tokens to compute token importance. However, our experiments show that the optimal choice of $n_{\text{window}}$ (the number of input queries used for importance estimation) is task-dependent, with some tasks benefiting from additional input queries. To address this, we adopt a large window with non-uniform token weights, placing greater emphasis on tokens near the prompt's end to achieve robust, task-wide performance. Window tokens are reweighted so that the $j$th token from the end receives weight $\frac{n_{\text{window}} - (j-1)}{n_{\text{window}}}$. Max aggregation is performed across layers, heads, and queries to produce a single importance score per token (excluding the always-kept last $n_{\text{window}}$ tokens). Additionally, as PyramidKV (Cai et al., 2024b) showed that attention is more focused in deeper layers, we exclude the first $l_{\text{skip}}$ layers during aggregation.

We apply average, then max pooling, so selected tokens also include nearby context, avoiding static chunking that could split related tokens (e.g., key-value pairs). This maintains the local context LLMs require. Unlike other methods that select entire sentences, we avoid sentence-level pre-processing to support non-text inputs, such as images. We then select the top-$C_{\text{max}}$ tokens with the highest scores, always including window tokens, to form the compressed prompt. We pass the compressed prompt to the target model with reassigned contiguous position IDs, enabling SpecPC to surpass the model's maximum context length.

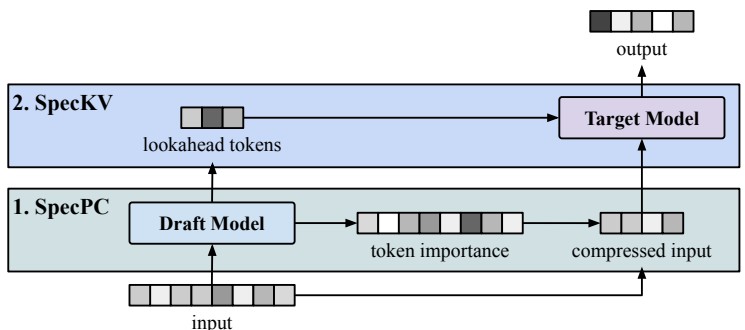

Figure 4: Overview of cascaded compression with SpecKV-PC: First, the draft model produces token importance scores and lookahead tokens. Next, SpecPC uses these scores to compress the initial input prompt. Finally, the target model is prefilled using both the compressed prompt and the lookahead tokens, while SpecKV compresses its KV cache

### 4.3 SPECKV-PC: CASCADED COMPRESSION WITH SPECKV AND SPECPC

SpecKV-PC integrates SpecKV and SpecPC into a highly efficient, two-stage compression pipeline. The core strategy leverages a cascaded approach, which first compresses the prompt with SpecPC and then further compresses the KV cache with SpecKV (Fig. 4). Because of fewer target model activations, SpecKV-PC achieves substantially lower latency and a smaller memory footprint than SpecKV alone, as the computationally intensive target model processes only a fraction of the original prompt.

## 5 EXPERIMENTS

### 5.1 SETUP

We benchmark SpecKV and SpecPC against baselines on RULER (Hsieh et al., 2024) and Long-Bench (Bai et al., 2024) using two model pairs: Qwen2.5-Instruct (Yang et al., 2024) (14B target with 1.5B/0.5B drafts for KV dropping/prompt compression) and Llama-3-Instruct (Grattafiori et al., 2024) (3.1-70B 4-bit target with 3.2-3B/1B drafts, respectively).

RULER is a synthetic benchmark with 13 tasks of varying complexity, including tasks such as key-value retrieval (NIAH), multi-hop tracing, and aggregation. It can be generated at any sequence length to assess a model's effective context window. We evaluate at 4K, 8K, 16K, 32K, and 64K (Qwen is excluded at 64K due to its 32K sequence limit). LongBench contains 12 English, five Chinese, two code, and three synthetic tasks across five categories. We exclude the Chinese tasks (unsupported by Llama) and synthetic tasks (already covered by RULER). We select 50 examples from each task, yielding 700 examples from LongBench and 650 examples at each context length for RULER.

For SpecKV, we compare against KV dropping methods H2O (Zhang et al., 2023), SnapKV (Li et al., 2024), PyramidKV (Cai et al., 2024b), and LAQ++ (Wang et al., 2025), using $C_{max} = 256$. Additionally, we assess SpecKV-PC-2048 that first compresses to 2048 tokens via SpecPC before applying SpecKV ($C_{max} = 256$). For SpecPC, we benchmark against LLMLingua-v2 (Pan et al., 2024), CPC (Liskavets et al., 2025), R2C (Choi et al., 2024), and SpecPrefill (Liu et al., 2025) with $C_{max} = 1024$. Based on our ablation studies (Fig. 19), we set $n_{lookahead}$ to the maximum token limit for SpecKV and to one for SpecPC. For SpecPrefill and LAQ++, we use $n_{lookahead} = 8$ as in the official paper. Sections C to E contain datasets, details, and additional results, including cross-family evaluations, different $C_{max}$ settings, and multimodal experiments on MileBench (Dingjie et al., 2024) with Qwen2.5-VL (Bai et al., 2025).

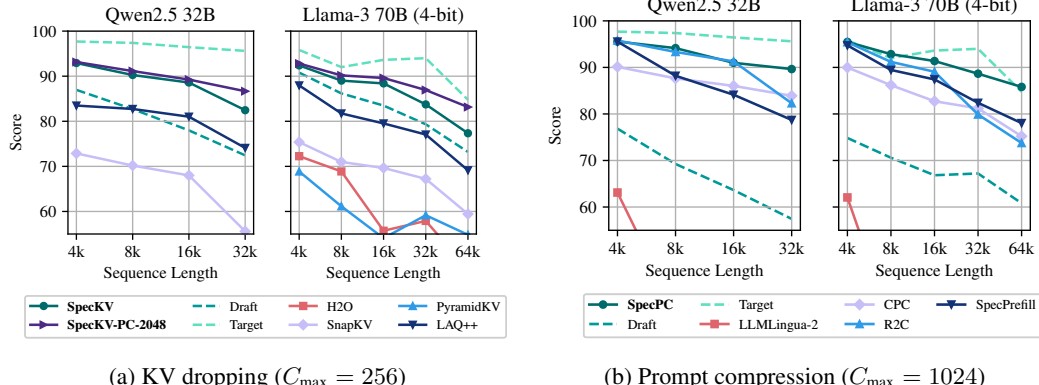

(a) KV dropping ($C_{\max} = 256$)  (b) Prompt compression ($C_{\max} = 1024$)

Figure 5: Performance of SpecKV and SpecPC. Both methods consistently outperform all baselines across sequence lengths, maintaining strong results at longer contexts. SpecKV-PC further improves upon SpecKV to achieve state-of-the-art results for KV dropping. Note that H2O and PyramidKV are not plotted for Qwen2.5 32B as their performance falls outside the visible range.

## 5.2 RESULTS

Fig. 5 (RULER) and Table 2 (LongBench) compare our methods with baselines. All our methods consistently outperform other baselines, demonstrating superior KV cache and prompt compression. Their performance far exceeds the draft model, highlighting robustness even with weaker drafts. Performance improves further with better drafts (Fig. 18). On RULER and LongBench, SpecKV and SpecPC consistently surpass existing baselines across multiple model families and context lengths. Interestingly, SpecKV-PC outperforms SpecKV alone, suggesting that the initial SpecPC-based prompt compression acts as an effective pre-filter by removing easy-to-identify, unimportant tokens. For larger $C_{\max}$, our methods remain superior (Figs. 10 and 11; Tables 9 to 11). Finally, we provide additional results regarding AdaKV (Section E.2), multimodal tasks (Section E.3), cross-family settings (Section E.5), and SpecKV-PC prompt compression ratios (Section E.6).

Table 2: LongBench performance with Qwen2.5 and Llama-3.

| | | Qwen2.5 32B | | | | | | Llama-3 70B (4-bit) | | | | | |
| Group | Method | SingleQA | MultiQA | Summ. | Few-shot | Code | All | SingleQA | MultiQA | Summ. | Few-shot | Code | All |
|---|---|---|---|---|---|---|---|---|---|---|---|---|---|
| Dense | Target | 56.01 | 43.99 | 25.90 | 64.06 | 44.74 | 47.78 | 55.02 | 47.06 | 28.61 | 70.47 | 48.19 | 49.99 |
| KV | H2O | 46.63 | 30.81 | 19.88 | 56.03 | 39.27 | 39.32 | 54.07 | 41.30 | 22.55 | 49.10 | 54.14 | 43.52 |
| | SnapKV | 52.54 | 40.21 | 19.89 | 61.18 | 40.12 | 42.98 | _55.88_ | 45.30 | 22.49 | 62.15 | 55.49 | 47.75 |
| | PyramidKV | 50.92 | 37.26 | 18.90 | 63.24 | 40.20 | 43.19 | 55.41 | 45.59 | 22.50 | 59.06 | 49.90 | 46.25 |
| | LAQ++ | **55.15** | _44.14_ | 22.24 | 63.25 | 41.19 | 45.79 | 54.90 | 46.48 | 22.83 | _64.31_ | 55.10 | 48.43 |
| | **SpecKV** | _53.48_ | 43.77 | _24.02_ | **63.79** | _44.80_ | _46.06_ | 51.80 | **47.23** | _25.53_ | 64.02 | **58.75** | _48.80_ |
| | **SpecKV-PC-2048** | 52.60 | **44.52** | **24.11** | _63.38_ | **48.45** | **46.48** | **61.42** | _47.15_ | **26.51** | **66.94** | _58.19_ | **51.60** |
| PC | LLMLingua-2 | 33.83 | 26.39 | 22.85 | 32.46 | _43.01_ | 30.90 | 37.95 | 28.20 | 23.35 | 42.37 | 37.63 | 33.63 |
| | CPC | 45.60 | 40.62 | 23.09 | 60.08 | 32.31 | 40.91 | 45.14 | 39.41 | 24.86 | 61.40 | 37.58 | 41.97 |
| | R2C | 50.49 | 40.37 | _23.26_ | 53.45 | 34.11 | 39.88 | 48.93 | 42.01 | 25.38 | 58.91 | 40.19 | 43.29 |
| | SpecPrefill | 45.94 | 39.32 | 23.16 | _62.04_ | **43.17** | _42.70_ | _54.62_ | **46.43** | _25.63_ | _64.80_ | _44.92_ | _48.37_ |
| | **SpecPC** | 51.23 | 41.40 | **23.37** | **62.26** | 38.23 | **43.66** | **56.84** | _44.48_ | **25.91** | **67.37** | **47.15** | **48.44** |

## 5.3 EFFICIENCY

We evaluate latency and memory on a single NVIDIA H200 (141GB) GPU. The target model is Qwen2.5-32B, with draft models of Qwen2.5-1.5B for KV dropping and Qwen2.5-0.5B for prompt compression. Latency is measured as the time to generate 64 tokens including all draft stages, with $n_{\text{lookahead}}$ set to 64 for SpecKV and SpecKV-PC[4], 8 for SpecPrefill and LAQ++, and 1 for SpecPC, with SpecKV-PC prompts compressed to 2048 tokens. For KV dropping methods, we report peak system memory, while for prompt compression, we report memory for the compression stage only,

---

[4]While some tasks can generate up to 128 tokens, benchmark outputs are typically under 64 tokens.

because the target model uses the same amount of memory for all prompt compression algorithms. A detailed breakdown of latency and memory is included in Section E.7.

SpecKV outperforms both SnapKV and LAQ++ in speed, driven by efficient sparse prefilling and a low-latency draft model (Fig. 6). Notably, SpecKV-PC achieves a 75% reduction in latency compared to LAQ++ at 64k context, as it significantly reduces the target prefill bottleneck through prompt compression. Regarding memory, while SpecKV requires more memory than SnapKV to store draft weights, this overhead is constant and can be further reduced by offloading to a CPU. Crucially, SpecKV is far more memory-efficient than LAQ++, which needs the entire target KV cache to function, matching the memory footprint of the dense target model. Thus, SpecKV offers a superior combination of accuracy and memory efficiency. Furthermore, SpecKV-PC yields substantial peak memory savings (around 25GB compared to LAQ++ at 64k context) by feeding a shorter prompt to the target model. Other KV dropping methods are omitted as their performance is similar to SnapKV.

SpecPC achieves the lowest latency among all baselines. It avoids the CPU preprocessing overhead of CPC and R2C and is faster than SpecPrefill due to a shorter lookahead. SpecPC is also the most memory-efficient, using substantially less memory than R2C. Overall, prompt compression is faster than KV dropping because only the compressed prompt tokens ($C_{\max}$) are passed to the target model.

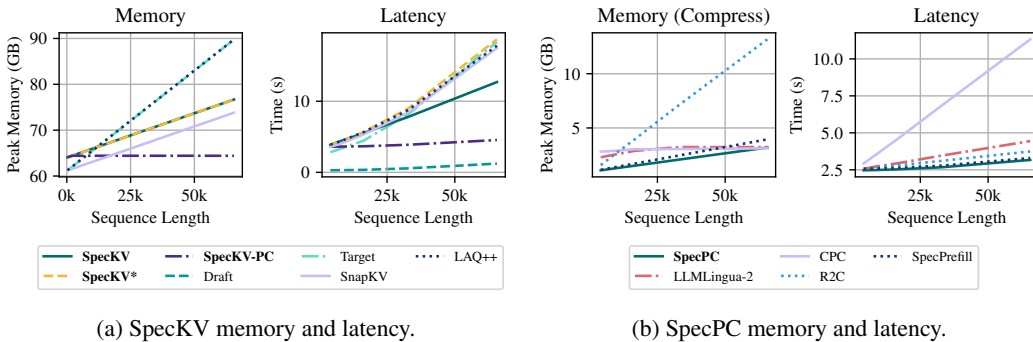

(a) SpecKV memory and latency.  (b) SpecPC memory and latency.

Figure 6: Peak memory usage and latency. SpecKV* denotes SpecKV without sparse prefill.

## 6 DISCUSSION

In this paper, we present Draft-based Approximate Inference, a framework that leverages draft model lookahead for approximate inference. Within this framework, we introduce two concrete algorithms: Speculative KV Dropping (SpecKV) for KV cache dropping with sparse prefill, and Speculative Prompt Compression (SpecPC) for prompt compression. We further propose SpecKV-PC, a cascaded pipeline that synergizes these approaches for improved efficiency and accuracy. Our approach is grounded in theoretical and empirical analyses that justify lookahead-based KV cache dropping and the use of draft model token importance to approximate target model importance. Through comprehensive experiments on long-context benchmarks, we show that our methods consistently achieve state-of-the-art accuracy under fixed KV cache and prompt size constraints, surpassing prior baselines. These contributions establish draft model lookahead as a powerful tool for efficient long-context inference, extending the role of draft models beyond speculative decoding.

**Limitations and Future Work**  For SpecKV, draft generation causes minimal latency even for larger $n_{\mathrm{lookahead}}$ (Section E.7). However, very long outputs or large $n_{\mathrm{lookahead}}$ values may reduce performance. In these cases, lowering $n_{\mathrm{lookahead}}$ could maintain speed with little loss in accuracy. For SpecPC, increasing $n_{\mathrm{lookahead}}$ to generate more tokens led to only minor accuracy gains; better leveraging longer drafts remains future work. Additionally, while our methods are robust using very small (Fig. 18) or cross-family (Section E.5) draft models, a reasonably accurate draft model is still required. Currently, Draft-based Approximate Inference supports sparse prefill, KV dropping, and prompt compression. Extensions such as lookahead-based sparse decoding or iterative KV cache dropping, where KV entries are periodically removed using draft lookahead, could further improve support for reasoning models with long outputs.

REPRODUCIBILITY STATEMENT

We provide a link to our public code. All algorithms, datasets, and experimental details, including hyperparameter settings, can be found in Section 5 and the appendix (Sections A, C and D).

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

# Appendix

# A    ALGORITHM PSEUDOCODE

---

**Algorithm 1** SpecKV

---

1: **Input:**

   Input sequence $x$ with length $n_{\text{in}}$

   Parameters: Number of lookahead tokens $n_{\text{lookahead}}$, Maximum cache capacity $C_{\text{max}}$,

   Compression window size $n_{\text{window}}$, Kernel size $k$, Prefill window size $n_{\text{slash}}$,

   Number of global tokens in prefill $n_{\text{vert}}$

2: Generate a draft output $y_{\text{draft}}$ of length $n_{\text{lookahead}}$ using the draft model.

3: Forward $x$ and $y_{\text{draft}}$ through the target model.

4: **for** each attention head in target model **do**

5:     $X \leftarrow$ Target model hidden states from prompt at current layer                    $\triangleright X \in \mathbb{R}^{n_{\text{in}} \times d}$

6:     $Y \leftarrow$ Target model hidden states from draft output at current layer            $\triangleright Y \in \mathbb{R}^{n_{\text{lookahead}} \times d}$

7:     $m \leftarrow n_{\text{in}} - n_{\text{window}}$

8:     $A \leftarrow \text{CrossAttention}\left(Q \text{ from } \left[\begin{smallmatrix} X_{m:} \\ Y \end{smallmatrix}\right],\ K/V \text{ from } X_{:m}\right)$          $\triangleright$ Compute attention

9:     $s \leftarrow \text{MaxReduce}(A)$                                                       $\triangleright \mathbb{R}^{n_1 \times n_2} \to \mathbb{R}^{n_2}$

10:    $s \leftarrow \text{AvgPool1D}(s, k)$                                                    $\triangleright$ Smooth attention

11:    $i_{\text{vert}} \leftarrow \text{topk}(s, n_{\text{vert}})$                             $\triangleright$ Select global tokens (Sparse prefill)

12:    $i_{\text{slash}} \leftarrow \{1, 2, \ldots, n_{\text{slash}}\}$                         $\triangleright$ Sliding window (Sparse prefill)

13:    $i_{\text{cache}} \leftarrow \text{topk}(s, C_{\text{max}} - n_{\text{window}}) \cup \{m+1, m+2, \ldots, n_{\text{in}}\}$          $\triangleright$ Select KVs

14:    $\text{output} \leftarrow \text{VerticalSlash}(X, i_{\text{vert}}, i_{\text{slash}})$    $\triangleright$ Sparse prefill attention

15:    $\text{cache} \leftarrow K_{i_{\text{cache}}}, V_{i_{\text{cache}}}$                     $\triangleright$ KV cache dropping

16: **end for**

---

**Algorithm 2** SpecPC

---

1: **Input:**

   Draft attention tensor $A \in \mathbb{R}^{n_{\text{layer}} \times n_{\text{head}} \times (n_{\text{in}} + n_{\text{lookahead}} - 1) \times n_{\text{in}}}$

   Parameters: Window size $n_{\text{window}}$, Kernel size $k$, Number of neighbors $n_{\text{neighbor}}$,

   Number of selected tokens $C_{\text{max}}$, Number of skipped layers $l_{\text{skip}}$

2: $m \leftarrow n_{\text{in}} - n_{\text{window}}$

3: $A \leftarrow A_{l_{\text{skip}}:, :, m:, :m}$                       $\triangleright$ Skip layers and only consider window queries and non-window keys

4: **for** $j \in \{1, 2, \ldots, n_{\text{window}}\}$ **do**

5:     $A_{\ldots, j, :} \leftarrow \frac{j}{n_{\text{window}}} A_{\ldots, j, :}$          $\triangleright$ Assign more weight to later tokens

6: **end for**

7: $s \leftarrow \text{MaxReduce}(A)$                                                       $\triangleright \mathbb{R}^{n_1 \times n_2 \times n_3 \times n_4} \to \mathbb{R}^{n_4}$

8: $s \leftarrow \text{AvgPool1D}(s, k)$                                                    $\triangleright$ Smooth attention

9: $s \leftarrow \text{MaxPool1D}(s, n_{\text{neighbor}})$                                  $\triangleright$ Keep neighbor tokens

10: $i_{\text{selected}} \leftarrow \text{topk}(s, C_{\text{max}}) \cup \{m+1, m+2, \ldots, n_{\text{in}}\}$          $\triangleright$ Keep most activated tokens and window tokens

11: **return** $i_{\text{selected}}$

---

# B    MATHEMATICAL PROOFS

**Lemma 1.** $\| \text{Softmax}(x) - \text{Softmax}(y) \|_2 \leq \| x - y \|_\infty$.

*Proof.* Let $J$ be the Jacobian matrix of Softmax. Then, $J(v) = \text{diag}(p) - pp^T$, where $p = \text{Softmax}(v)$. Note that $p$ is a probability distribution, so $p_i \geq 0$ for all $i$ and $\sum_i p_i = 1$. For any

vectors $v$ and $z$,

$$
\begin{aligned}
\|J(v)z\|_2^2 &= \|(\mathrm{diag}(p) - pp^T)z\|_2^2 \\
&= \sum_i (p_i z_i - p_i p^T z)^2 \\
&= \sum_i p_i^2 (z_i - p^T z)^2 \\
&\leq \sum_i p_i (z_i - p^T z)^2 \\
&= \sum_i \left( p_i z_i^2 - 2 p_i z_i p^T z + p_i (p^T z)^2 \right) \\
&= \sum_i p_i z_i^2 - (p^T z)^2 \\
&\leq \sum_i p_i z_i^2 \\
&\leq \sum_i p_i \|z\|_\infty^2 \\
&\leq \|z\|_\infty^2.
\end{aligned}
$$

Thus, $\|J(v)z\|_2 \leq \|z\|_\infty$ for all $v$ and $z$. From the fundamental theorem of line integrals,

$$
\mathrm{Softmax}(x) - \mathrm{Softmax}(y) = \int_0^1 J(y + t(x - y))(x - y)dt. \tag{4}
$$

Finally,

$$
\begin{aligned}
\|\mathrm{Softmax}(x) - \mathrm{Softmax}(y)\|_2 &= \left\| \int_0^1 J(y + t(x - y))(x - y)dt \right\|_2 \\
&\leq \int_0^1 \|J(y + t(x - y))(x - y)\|_2 dt \\
&\leq \int_0^1 \|x - y\|_\infty dt \\
&= \|x - y\|_\infty.
\end{aligned}
$$

$\square$

**Lemma 2.** *Let $y = \mathrm{Softmax}(x)$ and $y' = \mathrm{Softmax}(x')$. If $\|y - y'\|_p \leq \epsilon$, then there exists a scalar $c$ such that $\|x - x' - c\mathbf{1}\|_p \leq \frac{\epsilon}{m}$, where $m = \min_i(\min(y_i, y_i'))$ and $p \in \{1, 2, \ldots, \infty\}$.*

*Proof.* From the mean value theorem, there exists $\xi \in (y_i, y_i')$ such that

$$
\frac{\log y_i - \log y_i'}{y_i - y_i'} = \left. \frac{d \log t}{dt} \right|_{t=\xi} = \frac{1}{\xi}. \tag{5}
$$

Note that $\xi > 0$. Then,

$$
|\log y_i - \log y_i'| = \frac{1}{\xi}|y_i - y_i'| \leq \frac{1}{m}|y_i - y_i'|. \tag{6}
$$

Let $c = \log \sum_j e^{x_j} - \log \sum_j e^{x_j'}$, so

$$
\left| \log \frac{e^{x_i}}{\sum_j e^{x_j}} - \log \frac{e^{x_i'}}{\sum_j e^{x_j'}} \right| = \left| x_i - x_i' - \left( \log \sum_j e^{x_j} - \log \sum_j e^{x_j'} \right) \right| = |x_i - x_i' - c| \tag{7}
$$

for all $i$. Thus,

$$
\begin{aligned}
|x_i - x_i' - c| \leq \tfrac{1}{m}|y_i - y_i'| &\implies |x_i - x_i' - c|^p \leq \tfrac{1}{m^p}|y_i - y_i'|^p \\
&\implies \sum_i |x_i - x_i' - c|^p \leq \tfrac{1}{m^p}\sum_i |y_i - y_i'|^p \\
&\implies \|x - x' - c\mathbf{1}\|_p^p \leq \tfrac{1}{m^p}\|y - y'\|_p^p \\
&\implies \|x - x' - c\mathbf{1}\|_p \leq \tfrac{1}{m}\|y - y'\|_p \\
&\implies \|x - x' - c\mathbf{1}\|_p \leq \tfrac{\epsilon}{m}.
\end{aligned}
$$

$\square$

### B.1 PROOF OF THEOREM 1

We define the vector of importance scores as and its approximation as

$$
s^T = \tfrac{1}{n_{\text{out}}} \sum_{i=1}^{n_{\text{out}}} \text{Softmax}\left(\frac{x_i^{(o)T} W_q W_k^T X^T}{\sqrt{d}}\right) \quad \text{and} \quad \hat{s}^T = \tfrac{1}{n_{\text{out}}} \sum_{i=1}^{n_{\text{out}}} \text{Softmax}\left(\frac{\hat{x}_i^{(o)T} W_q W_k^T X^T}{\sqrt{d}}\right),
\tag{8}
$$

where $X = [x_1, \ldots, x_{n_{\text{in}}}]^T \in \mathbb{R}^{n_{\text{in}} \times d}$ is the matrix of input embeddings, $x_i^{(o)} \in \mathbb{R}^d$ is the $i$th output embedding, and $\hat{x}_i^{(o)} \in \mathbb{R}^d$ is the $i$th approximate output embedding (from the draft model). $s_i$ and $\hat{s}_i$ denote the importance of the $i$th KV pair. In practice, SpecKV estimates importance using queries from recent input and draft output tokens. This is omitted from the theoretical analysis for clarity.

*Proof.* We assume $\|x_i^{(o)} - \hat{x}_i^{(o)}\|_2 \leq \epsilon$ for all $i$ and $\|x_j\|_2 \leq \sqrt{d}$ for all $j$.

$\|x_i^{(o)} - \hat{x}_i^{(o)}\|_2 \leq \epsilon$, so

$$
\begin{aligned}
\left|\frac{x_i^{(o)T} W_q W_k^T x_j}{\sqrt{d}} - \frac{\hat{x}_i^{(o)T} W_q W_k^T x_j}{\sqrt{d}}\right| &= \tfrac{1}{\sqrt{d}}\left|(x_i^{(o)} - \hat{x}_i^{(o)})^T W_q W_k^T x_j\right| \\
&\leq \tfrac{1}{\sqrt{d}}\|W_q W_k^T\|_2 \epsilon \sqrt{d} \\
&= \epsilon\|W_q W_k^T\|_2.
\end{aligned}
$$

Thus,

$$
\left\|\frac{x_i^{(o)T} W_q W_k^T X^T}{\sqrt{d}} - \frac{\hat{x}_i^{(o)T} W_q W_k^T X^T}{\sqrt{d}}\right\|_\infty \leq \epsilon\|W_q W_k^T\|_2.
\tag{9}
$$

Applying Lemma 1 and the triangle inequality, we get

$$
\begin{aligned}
\|s - \hat{s}\|_2 &= \left\|\tfrac{1}{n_{\text{out}}} \sum_{i=1}^{n_{\text{out}}} \text{Softmax}\left(\frac{x_i^{(o)T} W_q W_k^T X^T}{\sqrt{d}}\right) - \tfrac{1}{n_{\text{out}}} \sum_{i=1}^{n_{\text{out}}} \text{Softmax}\left(\frac{\hat{x}_i^{(o)T} W_q W_k^T X^T}{\sqrt{d}}\right)\right\|_2 \\
&\leq \tfrac{1}{n_{\text{out}}} \sum_{i=1}^{n_{\text{out}}} \left\|\text{Softmax}\left(\frac{x_i^{(o)T} W_q W_k^T X^T}{\sqrt{d}}\right) - \text{Softmax}\left(\frac{\hat{x}_i^{(o)T} W_q W_k^T X^T}{\sqrt{d}}\right)\right\|_2 \\
&\leq \tfrac{1}{n_{\text{out}}} \sum_{i=1}^{n_{\text{out}}} \left\|\frac{x_i^{(o)T} W_q W_k^T X^T}{\sqrt{d}} - \frac{\hat{x}_i^{(o)T} W_q W_k^T X^T}{\sqrt{d}}\right\|_\infty \\
&\leq \epsilon\|W_q W_k^T\|_2,
\end{aligned}
$$

$\square$

### B.2 PROOF OF THEOREM 2

$$
X = [x_1, \ldots, x_{n_{\text{in}}}]^T
\tag{10}
$$

$$
Y = [y_1, \ldots, y_{n_{\text{in}}}]^T = \text{Softmax}\left(\frac{X W_q W_k^T X^T}{\sqrt{d}}\right) X W_v
\tag{11}
$$

$$\hat{Y} = [\hat{y}_1, \ldots, \hat{y}_{n_{\text{in}}}]^T = \text{Softmax}\left(\frac{X\hat{W}_q\hat{W}_k^T X^T}{\sqrt{\hat{d}}}\right) X\hat{W}_v \tag{12}$$

$$A = [a_1, \ldots, a_{n_{\text{in}}}]^T = \text{Softmax}\left(\frac{XW_q W_k^T X^T}{\sqrt{d}}\right) \tag{13}$$

$$\hat{A} = [\hat{a}_1, \ldots, \hat{a}_{n_{\text{in}}}]^T = \text{Softmax}\left(\frac{X\hat{W}_q\hat{W}_k^T X^T}{\sqrt{d}}\right) \tag{14}$$

*Proof.* We assume $\|y_i - \hat{y}_i\|_2 \le \epsilon\|X\|_{\infty,2}$, where $\|X\|_{\infty,2}$ is the maximum $\ell_2$ norm of the rows of $X$. Additionally, we assume that there exists a constant $c$ such that $cX^T$ has the Restricted Isometry Property (Candes & Tao, 2005) with parameters $(2k, \delta)$, where $\delta$ is the restricted isometry constant and $k$ is the approximate sparsity of $a_i$ and $\hat{a}_i$.

Recall that a matrix $B$ satisfies the Restricted Isometry Property with constant $\delta \in (0, 1)$ if for every $k$-sparse vector $v$, the following inequality holds:

$$(1 - \delta)\|v\|_2^2 \le \|Bv\|_2^2 \le (1 + \delta)\|v\|_2^2. \tag{15}$$

Let $\Delta W_v = W_v - \hat{W}_v$ and $\Delta a_i = a_i - \hat{a}_i$.

If $n_{\text{in}} = 1$, then $A = \hat{A} \in \mathbb{R}^{1 \times 1}$ with $A_{1,1} = \hat{A}_{1,1} = 1$, so $AX = \hat{A}X = X = x_1^T$, which implies

$$\|y_1 - \hat{y}_1\|_2 = \|x_1^T W_V - x_1^T \hat{W}_v\|_2 = \|x_1^T \Delta W_v\|_2 \le \epsilon\|X\|_{\infty,2} = \epsilon\|x_1\|_2 \tag{16}$$

for all $x_1$. $\|x_1^T W_v\|_2 \le \epsilon\|x_1\|$ for all $x_1$ is the definition of the matrix $\ell_2$ norm, so $\|\Delta W_v\|_2 \le \epsilon$.

$$
\begin{aligned}
\|y_i - \hat{y}_i\|_2 &= \|a_i^T X W_v - \hat{a}_i^T X \hat{W}_v\|_2 \\
&= \|a_i^T X W_v - (\hat{a}_i^T X W_v - \hat{a}_i^T X \Delta W_v)\|_2 \\
&= \|a_i^T X W_v - (a_i^T X W_v - \Delta a_i^T X W_v - \hat{a}_i^T X \Delta W_v)\|_2 \\
&= \|\Delta a_i^T X W_v + \hat{a}_i^T X \Delta W_v\|_2 \\
&\le \epsilon\|X\|_{\infty,2}
\end{aligned}
$$

Then, $\|\Delta a_i^T X W_v\|_2 \le \epsilon\|X\|_{\infty,2} + \|\hat{a}_i^T X \Delta W_v\|_2$.

Since $\hat{a}_i^T X$ is a convex combination of the rows of $X$, $\|\hat{a}_i^T X\|_2 \le \|X\|_{\infty,2}$.

Thus, $\|\Delta a_i^T X W_v\|_2 \le \epsilon\|X\|_{\infty,2} + \|\hat{a}_i^T X \Delta W_v\|_2 \le \epsilon\|X\|_{\infty,2} + \|\hat{a}_i^T X\|_2\|\Delta W_v\|_2 \le 2\epsilon\|X\|_{\infty,2}$.

Attention scores are approximately sparse (Jiang et al., 2024a), especially for long sequences. Therefore, we assume $a_i$ and $\hat{a}_i$ are $k$-sparse. Then, $\Delta a_i$ is at most $2k$-sparse. Since $cX^T$ has the Restricted Isometry Property with parameters $2k, \delta$,

$$(1 - \delta)\|\Delta a_i\|_2 \le \|\Delta a_i^T(cX)\|_2 \le (1 + \delta)\|\Delta a_i\|_2. \tag{17}$$

Then,

$$\tfrac{1}{c}(1 - \delta)\|\Delta a_i\|_2 \le \|\Delta a_i^T X\|_2 \le \frac{2\epsilon\|X\|_{\infty,2}}{\sigma_{\min}(W_v)}, \tag{18}$$

so

$$\|\Delta a_i\|_2 \le \frac{2c\epsilon\|X\|_{\infty,2}}{\sigma_{\min}(W_v)(1-\delta)}. \tag{19}$$

$\square$

### B.3 PROOF OF THEOREM 3

**Theorem 3.** *If $\|Y - \hat{Y}\|_2 \le \epsilon\|X\|_2$ for all $X$ and the column space of $W_q, W_k, \hat{W}_q, \hat{W}_k$ is a subset of the column space of $W_v$, then $\|a_i - \hat{a}_i\|_2 \le \epsilon\delta$, where*

$$\delta = 2d\frac{\sigma_{\max}(W_v)^2}{\sigma_{\min}(W_v)}\exp\left(2\max\left(\frac{\|W_q\|_2\|W_k\|_2}{\sigma_{\min}(W_v)^2}, \frac{\|\hat{W}_q\|_2\|\hat{W}_k\|_2}{\sigma_{\min}(\hat{W}_v)^2}\right)\right)\|X\|_{\infty,2}^2. \tag{20}$$

*Proof.* We assume $\|Y - \hat{Y}\|_2 \leq \epsilon\|X\|_2$. Additionally, we assume that the column space of $W_q, W_k, \hat{W}_q, \hat{W}_k$ is a subset of the column space of $W_v$. To get a norm bound on $\Delta a_i = a_i - \hat{a}_i$, we will bound the norms of the error in approximate weight matrices. We will find these bounds by using specific inputs, taking advantage of the fact that $\|Y - \hat{Y}\|_2 \leq \epsilon\|X\|_2$ for all $X$.

We will start by bounding $\Delta W_v = W_v - \hat{W}_v$, by choosing an input that fixes $A$ and $\hat{A}$. If $n = 1$, then $A = \hat{A} = [1]$, so $AX = \hat{A}X = X = x_1^T$, which implies

$$\|Y - \hat{Y}\|_2 = \|AXW_V - \hat{A}X\hat{W}_v\|_2 = \|x_1^T W_V - x_1^T \hat{W}_v\|_2 = \|x_1^T \Delta W_v\|_2 \leq \epsilon\|X\|_2 = \epsilon\|x_1\|_2 \tag{21}$$

for all $x_1$. Thus, $\|\Delta W_v\|_2 \leq \epsilon$.

Next, we will bound the norm of $\Delta B = B - \hat{B}$, where $B = W_q W_k^T$ and $\hat{B} = \hat{W}_q \hat{W}_k^T$. We will choose the $X$ so that the values are the identity matrix. Then $Y = A$. Let $U\Sigma V^T$ be the singular value decomposition of $W_v$. We set

$$X = \Phi V \Sigma^{-1} U^T, \tag{22}$$

where $\Phi$ is an arbitrary orthonormal basis spanning $\mathbb{R}^{d \times d}$.

Note that $\sigma_{\min}(\Phi) = \sigma_{\max}(\Phi) = 1$, so $\|X\|_2 = \sigma_{\max}(X) = \frac{1}{\sigma_{\min}(W_v)}$ and $\sigma_{\min}(X) = \frac{1}{\sigma_{\max}(W_v)}$.

Now,

$$
\begin{aligned}
\|Y - \hat{Y}\|_2 &= \|AXW_V - \hat{A}X\hat{W}_v\|_2 \\
&= \|AXW_V - (\hat{A}XW_v - \hat{A}X\Delta W_v)\|_2 \\
&= \|AXW_V - (AXW_v - \Delta AXW_v - \hat{A}X\Delta W_v)\|_2 \\
&= \|\Delta AXW_v + \hat{A}X\Delta W_v\|_2 \\
&\leq \|\Delta AXW_v\|_2 + \|\hat{A}X\Delta W_v\|_2 \\
&= \|\Delta A\Phi V\Sigma^{-1}U^T U\Sigma V^T\|_2 + \|\hat{A}X\Delta W_v\|_2 \\
&= \|\Delta A\Phi\|_2 + \|\hat{A}X\Delta W_v\|_2 \\
&= \|\Delta A\|_2 + \|\hat{A}X\Delta W_v\|_2 \\
&\leq \epsilon\|X\|_2 + \epsilon\|X\|_2 \\
&\leq \frac{2\epsilon}{\sigma_{\min}(W_v)}.
\end{aligned}
$$

Note that each row of $\hat{A}$ is a probability distribution (non-negative entries summing to 1), so left-multiplying $X$ by $\hat{A}$ forms a convex combination of the rows of $X$. From Jensen's inequality we get $\|\hat{A}X\|_2 \leq \|X\|_2$, because $x \to \|x\|_2$ is a convex function.

Let $\delta_1 = \max\left(\frac{\|W_q\|_2\|W_k\|_2}{\sigma_{\min}(W_v)^2}, \frac{\|\hat{W}_q\|_2\|\hat{W}_k\|_2}{\sigma_{\min}(\hat{W}_v)^2}\right)$. Since $\|X\|_2 = \frac{1}{\sigma_{\min}(W_v)}$ and $\|B\|_2 \leq \|W_q\|_2\|W_k\|_2$, $\|XBX^T\|_2 \leq \delta_1$. Consequently, $|x_i^T B x_j| \leq \delta_1$ for all $i, j$. This implies that each attention weight satisfies

$$a_{i,j} > \frac{e^{-\delta_1}}{\sum_{j=1}^d e^{\delta_1}} = \frac{1}{d}e^{-2\delta_1}. \tag{23}$$

The same argument applied to $\hat{a}$ gives

$$\hat{a}_{i,j} > \frac{e^{-\delta_1}}{\sum_{j=1}^d e^{\delta_1}} = \frac{1}{d}e^{-2\delta_1}. \tag{24}$$

Applying Lemma 2 to each row, there exists $c \in \mathbb{R}^d$ such that

$$\left\|\tfrac{XBX^T}{\sqrt{d}} - \tfrac{X\hat{B}X^T}{\sqrt{d}} + c\mathbf{1}^T\right\|_2 \le de^{2\delta_1}\left\|\text{Softmax}\left(\tfrac{XBX^T}{\sqrt{d}}\right) - \text{Softmax}\left(\tfrac{X\hat{B}X^T}{\sqrt{d}}\right)\right\|_2$$

$$\left\|XBX^T - X\hat{B}X^T + \sqrt{d}c\mathbf{1}^T\right\|_2 \le d^{3/2}e^{2\delta_1}\|\Delta A\|_2 \tag{25}$$

$$\le \tfrac{2\epsilon d^{3/2}e^{2\delta_1}}{\sigma_{\min}(W_v)}.$$

Minimizing over $c$, we obtain

$$\min_c \|XBX^T - X\hat{B}X^T + \sqrt{d}c\mathbf{1}^T\|_2 = \min_c \|X\Delta BX^T - \sqrt{d}c\mathbf{1}^T\|_2$$

$$= \left\|X\Delta BX^T - \tfrac{1}{d}X\Delta BX^T\mathbf{1}\mathbf{1}^T\right\|_2 \tag{26}$$

$$= \left\|X\Delta BX^T\left(I - \tfrac{1}{d}\mathbf{1}\mathbf{1}^T\right)\right\|_2$$

Substituting in the definition of $X$, we get

$$\|\Phi V\Sigma^{-1}U^T\Delta BU\Sigma^{-1}V^T\Phi^T(I - d^{-1}\mathbf{1}\mathbf{1}^T)\|_2 \le \tfrac{\epsilon d^{3/2}e^{2\delta_1}}{\sigma_{\min}(W_v)}. \tag{27}$$

Each multiplication by $\Sigma^{-1}$ can decrease the norm by at most $\frac{1}{\sigma_{\max}(W_v)}$, so when removing both instances of $\Sigma^{-1}$ we scale the bound by $\sigma_{\max}(W_v)^2$, giving us

$$\|\Phi VU^T\Delta BUV^T\Phi^T(I - d^{-1}\mathbf{1}\mathbf{1}^T)\|_2 \le \tfrac{2\epsilon d^{3/2}e^{2\delta_1}}{\sigma_{\min}(W_v)}\sigma_{\max}(W_v)^2. \tag{28}$$

Then, since $\Phi$ and $V$ are orthonormal and preserve spectral norm under multiplication, we conclude

$$\|U^T\Delta BUV^T\Phi^T(I - d^{-1}\mathbf{1}\mathbf{1}^T)\|_2 \le \tfrac{2\epsilon d^{3/2}e^{2\delta_1}}{\sigma_{\min}(W_v)}\sigma_{\max}(W_v)^2. \tag{29}$$

Finally, since the column space of $W_q, W_k, \hat{W}_q, \hat{W}_k$ is a subset of the column space of $W_v$, the column space of $\Delta B$ is a subset of the column space of $U$. Thus, left multiplication by $U^T$ does not impact the spectral norm, so

$$\|\Delta BUV^T\Phi^T(I - d^{-1}\mathbf{1}\mathbf{1}^T)\|_2 \le \tfrac{2\epsilon d^{3/2}e^{2\delta_1}}{\sigma_{\min}(W_v)}\sigma_{\max}(W_v)^2. \tag{30}$$

Note that the matrix $I - \tfrac{1}{d}\mathbf{1}\mathbf{1}^T$ is a projection onto the subspace orthogonal to the all-ones vector. Its singular values are $[1,\dots,1,0]$, so its spectral norm is

$$\left\|I - \tfrac{1}{d}\mathbf{1}\mathbf{1}^T\right\|_2 = 1. \tag{31}$$

Moreover, since $\left\|I - \tfrac{1}{h}\mathbf{1}\mathbf{1}^T\right\|_2 = 1$ and $\Phi$ is an arbitrary orthonormal basis of $\mathbb{R}^d$, it follows that for any fixed $P \in \mathbb{R}^{d\times d}$, we can choose $\Phi$ such that the largest component of $P\Phi^T$ lies entirely in the subspace orthogonal to $\mathbf{1}$. In this case,

$$\left\|P\Phi^T(I - \tfrac{1}{d}\mathbf{1}\mathbf{1}^T)\right\|_2 = \|P\|_2. \tag{32}$$

Thus, $\|\Delta B\|_2 \le \delta_2$ where $\delta_2 = 2\epsilon d^{3/2}\tfrac{\sigma_{\max}(W_v)^2}{\sigma_{\min}(W_v)}e^{2\delta_1}$.

Now that we have bounded $\|\Delta B\|_2$, we will consider any input $X$. Then, $|x_i^T\Delta Bx_j| \le \delta_2\|X\|_{\infty,2}^2$, so $\|x_i^T\Delta BX^T\|_\infty \le \delta_2\|X\|_{\infty,2}^2$. $\|X\|_{\infty,2}$ is the maximum $\ell_2$ norm of the rows of $X$.

From Lemma 1,

$$\|a_i - \hat{a}_i\|_2 = \|\,\text{Softmax}\left(\tfrac{x_i^T BX^T}{\sqrt{d}}\right) - \text{Softmax}\left(\tfrac{x_i^T\hat{B}X^T}{\sqrt{d}}\right)\|_2$$

$$\le \left\|\tfrac{x_i^T BX^T}{\sqrt{d}} - \tfrac{x_i^T\hat{B}X^T}{\sqrt{d}}\right\|_\infty = \tfrac{1}{\sqrt{d}}\left\|x_i^T\Delta BX^T\right\|_\infty \le \tfrac{\delta_2\|X\|_{\infty,2}^2}{\sqrt{d}}.$$

$$\|a_i - \hat{a}_i\|_2 \leq \frac{\delta_2 \|X\|_{\infty,2}^2}{\sqrt{d}} = 2\epsilon d \frac{\sigma_{\max}(W_v)^2}{\sigma_{\min}(W_v)} \exp\left(2\max\left(\frac{\|W_q\|_2 \|W_k\|_2}{\sigma_{\min}(W_v)^2}, \frac{\|\hat{W}_q\|_2 \|\hat{W}_k\|_2}{\sigma_{\min}(\hat{W}_v)^2}\right)\right) \|X\|_{\infty,2}^2. \tag{33}$$

$\square$

## C BENCHMARK DATASET DETAILS

### C.1 LONGBENCH

Table 3: LongBench tasks.

| Task | Dataset | Source | Avg. Words | Metric | Language | Size |
|------|---------|--------|-----------|--------|----------|------|
| **Single-Document QA** | | | | | | |
| 1-1 | NarrativeQA | Literature, Film | 18,409 | F1 | English | 200 |
| 1-2 | Qasper | Science | 3,619 | F1 | English | 200 |
| 1-3 | MultiFieldQA-en | Multi-field | 4,559 | F1 | English | 150 |
| 1-4 | MultiFieldQA-zh | Multi-field | 6,701 | F1 | Chinese | 200 |
| **Multi-Document QA** | | | | | | |
| 2-1 | HotpotQA | Wikipedia | 9,151 | F1 | English | 200 |
| 2-2 | 2WikiMultihopQA | Wikipedia | 4,887 | F1 | English | 200 |
| 2-3 | MuSiQue | Wikipedia | 11,214 | F1 | English | 200 |
| 2-4 | DuReader | Baidu Search | 15,768 | Rouge-L | Chinese | 200 |
| **Summarization** | | | | | | |
| 3-1 | GovReport | Government report | 8,734 | Rouge-L | English | 200 |
| 3-2 | QMSum | Meeting | 10,614 | Rouge-L | English | 200 |
| 3-3 | MultiNews | News | 2,113 | Rouge-L | English | 200 |
| 3-4 | VCSUM | Meeting | 15,380 | Rouge-L | Chinese | 200 |
| **Few-shot Learning** | | | | | | |
| 4-1 | TREC | Web question | 5,177 | Accuracy (CLS) | English | 200 |
| 4-2 | TriviaQA | Wikipedia, Web | 8,209 | F1 | English | 200 |
| 4-3 | SAMSum | Dialogue | 6,258 | Rouge-L | English | 200 |
| 4-4 | LSHT | News | 22,337 | Accuracy (CLS) | Chinese | 200 |
| **Synthetic Task** | | | | | | |
| 5-1 | PassageCount | Wikipedia | 11,141 | Accuracy (EM) | English | 200 |
| 5-2 | PassageRetrieval-en | Wikipedia | 9,289 | Accuracy (EM) | English | 200 |
| 5-3 | PassageRetrieval-zh | C4 Dataset | 6,745 | Accuracy (EM) | Chinese | 200 |
| **Code Completion** | | | | | | |
| 6-1 | LCC | Github | 1,235 | Edit Sim | Python/C#/Java | 500 |
| 6-2 | RepoBench-P | Github repository | 4,206 | Edit Sim | Python/Java | 500 |

LongBench[5] (Bai et al., 2024) is a benchmark suite designed for long-context evaluation, comprising 14 English tasks, five Chinese tasks, and two code tasks. As Llama does not support Chinese, we excluded the corresponding tasks. Furthermore, we removed the synthetic tasks, as these are already covered by the RULER benchmark. The remaining tasks are grouped into five categories: single-document question answering, multi-document question answering, summarization, few-shot learning, and code completion. For each category, the overall score is calculated as the average of all its subtasks. The final LongBench score is computed as the average across all included tasks. Table 3 provides an overview of all tasks, adapted from Bai et al. (2024).

### C.2 RULER

RULER[6] (Hsieh et al., 2024) is a synthetic dataset designed to evaluate the true supported context length of LLMs. It comprises 13 tasks, including eight needle-in-a-haystack (NIAH) retrieval tasks, two aggregation tasks, two question answering (QA) tasks, and one multi-hop tracing task.

The NIAH tasks involve hiding random key-value pairs within generated text and challenging the model to retrieve them. Aggregation tasks simulate summarization by asking the model to extract the

---

[5]`https://huggingface.co/datasets/THUDM/LongBench` (MIT License)
[6]`https://github.com/NVIDIA/RULER` (Apache License 2.0)

most frequent or common words from a given passage. The QA tasks require the model to answer a question about a randomly selected paragraph within the context, serving as a real-world analog to NIAH tasks. In the multi-hop tracing task, the model must identify all variable names that reference the same value within a chain of assignments.

RULER is generated for a range of sequence lengths using randomly generated texts drawn from Paul Graham essays, SQuAD (Rajpurkar et al., 2016), and HotPotQA (Yang et al., 2018) datasets. This approach enables a comprehensive assessment of a language model's capability to process varying context lengths. Evaluation is conducted based on accuracy, considering a response correct if it contains the requested value associated with the specified key.

## C.3 MILEBENCH

Table 4: Overview of the MileBench datasets. Average tokens are computed using Qwen2.5-VL (Bai et al., 2025).

| Category | Dataset | Avg. Words | Avg. Images | Avg. Tokens | Metric | Size |
|---|---|---|---|---|---|---|
| **Temporal** | EgocentricNavigation | 85 | 45 | 3,079 | Accuracy | 200 |
| | MovingDirection | 62 | 5 | 1,042 | Accuracy | 200 |
| | SceneTransition | 66 | 20 | 5,125 | Accuracy | 200 |
| **Semantic** | SlideVQA | 66 | 2 | 2,053 | Accuracy | 200 |
| | TQA | 50 | 8 | 5,536 | Accuracy | 200 |
| | WebQA | 146 | 2 | 1,706 | Accuracy | 200 |

MileBench[7] (Dingjie et al., 2024) is a long-context benchmark designed to evaluate Multimodal Large Language Models (MLLMs). It comprises 29 multi-image-text datasets, organized into 12 tasks, which are further grouped into four categories: Temporal Multi-Image, Semantic Multi-Image, and two diagnostic categories—NIAH and Image Retrieval.

For our additional experiments in Section E.3, we selected three datasets each from the Temporal Multi-Image and Semantic Multi-Image categories: EgocentricNavigation, MovingDirection, and SceneTransition for the Temporal Multi-Image category, and SlideVQA, TQA, and WebQA for the Semantic Multi-Image category. Table 4 provides an overview of the selected datasets.

## D EXPERIMENTAL SETUP

### D.1 HYPERPARAMETER SETTINGS

Table 5: Prompt compression backbones and parameter counts.

| Method | Backbone | Parameters |
|---|---|---|
| LLMLingua-2 (Pan et al., 2024) | xlm-roberta-large (Conneau et al., 2020) | 560M |
| CPC (Liskavets et al., 2025) | Llama-3.2-1B (Grattafiori et al., 2024) | 1B |
| R2C (Choi et al., 2024) | T5-base (Raffel et al., 2020) | 220M |

Table 6: Summary of hyperparameters for various methods.

| Hyperparameter | StreamingLLM | H2O | SnapKV | PyramidKV | Ada-SnapKV | LAQ++ | SpecKV | SpecPrefill | SpecPC |
|---|---|---|---|---|---|---|---|---|---|
| Window size $n_{\text{window}}$ | 32 | 32 | 32 | 32 | 32 | 32 | 32 | 1 | 64 |
| Pool | – | – | Max | Max | Max | Max | Max | Avg | Avg |
| Kernel size $k$ | – | – | 7 | 7 | 7 | 7 | 7 | 13 | 64/32 |
| Reduction | – | – | Mean | Mean | Mean | Max | Max | Mean-Max | Max |
| # lookahead tokens $n_{\text{lookahead}}$ | – | – | – | – | – | 8 | All | 8 | 1 |
| Compression window size $n_{\text{slash}}$ | – | – | – | – | – | – | 2048 | – | – |
| # global tokens in prefill $n_{\text{vert}}$ | – | – | – | – | – | – | 2048 | – | – |
| # neighbors $n_{\text{neighbor}}$ / chunk size | – | – | – | – | – | – | – | 32 | 64/32 |
| # skipped layers $l_{\text{skip}}$ | – | – | – | – | – | – | – | – | 8 |
| Initial cache size | – | – | – | – | – | $C_{\text{max}}$ | – | – | – |

---

[7] https://milebench.github.io (Apache License 2.0)

Table 5 lists the backbone models employed by each prompt compression method, while Table 6 details the hyperparameters used in our experiments. Generally, we select hyperparameters for each method based on their respective codebases. We observe that using max aggregation improved performance compared to mean aggregation for SpecKV and SpecPC. For SpecKV, setting $n_{\text{slash}}$ and $n_{\text{vert}}$ to 2048 resulted in minimal accuracy loss but substantially reduced latency (Section E.8.3).

For SpecKV, we always generate tokens until the draft model produces the EOS token, which yields the best performance. For latency measurements, we set $n_{\text{lookahead}} = 64$ tokens, reflecting the average sequence length in our benchmarks. In SpecPC, prompt compression drops tokens uniformly across all layers and heads (unlike SpecKV, which prunes per head), so a larger $C_{\text{max}}$ is needed to retain relevant information. While a larger $n_{\text{lookahead}}$ can boost performance, in practice, generating only one token per prompt ($n_{\text{lookahead}} = 1$) is usually sufficient. Strong alignment between the draft and target model attentions enables SpecPC to outperform methods like R2C and CPC. For an ablation on $n_{\text{lookahead}}$, see Fig. 19.

Retaining the local context for prompt compression proved essential. This observation aligns with the design of existing prompt compression methods, which typically aim to preserve entire sentences within the prompt. Consequently, we increase both the pooling kernel size ($k$) and the number of neighboring tokens ($n_{\text{neighbor}}$) to 64. For Llama, slightly better results are achieved by reducing both $k$ and $n_{\text{neighbor}}$ to 32, though the performance difference was marginal.

For all remaining methods not explicitly mentioned, we use the default configurations provided in their respective codebases.

### D.2 Implementation Details

For our experimental results, we employ the following large language models: **Llama-3.2-1B-Instruct**[8], **Llama-3.1-8B-Instruct**[9], **Llama-3.1-70B-Instruct (4-bit)**[10], **Qwen2.5-0.5B-Instruct**[11], **Qwen2.5-14B-Instruct**[12], **Qwen2.5-32B-Instruct**[13], and **Qwen2.5-72B-Instruct-GPTQ-Int4**[14]. For MLLM evaluation on MileBench (Dingjie et al., 2024), we utilize **Qwen2.5-VL-3B-Instruct-AWQ**[15] and **Qwen2.5-VL-32B-Instruct-AWQ**[16].

Our implementation is based on PyTorch (Paszke et al., 2017) (BSD-3 License) and Huggingface's Transformers (Wolf et al., 2019) (Apache License 2.0). All experiments leverage FlashAttention-2[17] (Dao, 2024). Latency measurements are performed using vLLM[18] wherever possible (i.e., where attention map outputs are not required). For implementing the sparse prefill mechanism of SpecKV, we use kernels from MInference[19]. All methods are evaluated via greedy decoding. Experiments are conducted on NVIDIA H100 80GB GPUs, with runtimes varying by context length; for a maximum context length of 64K tokens, experiments take up to 2 hours.

For evaluating StreamingLLM (Xiao et al., 2024), H2O (Zhang et al., 2023), SnapKV (Li et al., 2024), and PyramidKV (Cai et al., 2024b), we use implementations from KVCache-Factory[20]. In this library, the StreamingLLM and H2O implementations drop KV once after prefill, rather than at each decoding step, differing from their original codebases. This adjustment enables fairer comparison to SnapKV and others. We extend KVCache-Factory to support Grouped Query Attention (Ainslie et al., 2023) by repeating keys and values for each KV head, computing attention within the window, and averaging across KV heads. This approach avoids duplicating the KV cache.

---

[8] https://huggingface.co/meta-llama/Llama-3.2-1B-Instruct (Llama 3.2 license)

[9] https://huggingface.co/meta-llama/Llama-3.1-8B-Instruct (Llama 3.1 license)

[10] https://huggingface.co/meta-llama/Llama-3.1-70B-Instruct (Llama 3.1 license)

[11] https://huggingface.co/Qwen/Qwen2.5-0.5B-Instruct (Apache License 2.0)

[12] https://huggingface.co/Qwen/Qwen2.5-14B-Instruct (Apache License 2.0)

[13] https://huggingface.co/Qwen/Qwen2.5-32B-Instruct (Apache License 2.0)

[14] https://huggingface.co/Qwen/Qwen2.5-72B-Instruct-GPTQ-Int4 (Apache License 2.0)

[15] https://huggingface.co/Qwen/Qwen2.5-VL-3B-Instruct-AWQ (Apache License 2.0)

[16] https://huggingface.co/Qwen/Qwen2.5-VL-32B-Instruct-AWQ (Apache License 2.0)

[17] https://github.com/Dao-AILab/flash-attention (BSD 3-Clause License)

[18] https://github.com/vllm-project/vllm (Apache License 2.0)

[19] https://github.com/microsoft/MInference (MIT License)

[20] https://github.com/Zefan-Cai/KVCache-Factory (MIT License)

For other baselines, we use their official implementations. For Lookahead Q-Cache (LAQ++), we provide our own implementation since their code is not publicly available.

# E    EXTENDED EXPERIMENTAL ANALYSIS

## E.1    ADDITIONAL RESULTS ON RULER AND LONGBENCH

In this section, we present results for RULER and LongBench using various $C_{\max}$ values of 256, 512, and 1024 for KV dropping, and 1024, 2048, and 3072 for prompt compression. Specifically, we employ Qwen2.5-32B-Instruct, Llama-3.1-70B-Instruct (4-bit quantized with bitsandbytes[21]), and Qwen2.5-72B-Instruct-GPTQ-Int4 as target model.

Figs. 10 and 11 show the RULER results, and Tables 9 to 11 present the LongBench results. Overall, our methods achieve higher accuracy than the baselines in most settings, especially with small $C_{\max}$. Specifically, SpecKV significantly outperforms SnapKV and LAQ++ on RULER in most cases. Similarly, SpecPC consistently achieves strong results, particularly at longer sequence lengths on RULER. On LongBench, both of our methods also surpass the baselines.

## E.2    INTEGRATION AND COMPARISON WITH THE ADAKV BASELINE

Fig. 12 (RULER), Tables 12 and 13 (LongBench) present the performance of our proposed methods on Qwen2.5 (0.5B draft, 14B target) and Llama-3 (1B draft, 8B target). We include additional baselines, notably AdaKV (Feng et al., 2025), for various values of $C_{\max}$. AdaKV is an extension to SnapKV and allows different KV budget per attention head. We apply AdaKV to SpecKV in a similar fashion, which further boosts performance in our experiments.

## E.3    MULTI-MODAL EVALUATION

We conduct additional experiments using Qwen2.5-VL-3B-Instruct-AWQ (draft) and Qwen2.5-VL-32B-Instruct-AWQ (target) on six MileBench (Dingjie et al., 2024) datasets. We select three datasets each from the Temporal Multi-Image (EgocentricNavigation, MovingDirection, SceneTransition) and Semantic Multi-Image (SlideVQA, TQA, WebQA) categories. We focus on these datasets because, for Qwen2.5-VL, the performance gap between draft and target models is most significant; in other cases, the models perform too similarly or the draft even outperforms the target.

For KV dropping, we evaluate H2O, SnapKV, and PyramidKV from our prior experiments. We do not include AdaKV in our evaluation as it is dependent on an older Transformers (Wolf et al., 2019) version incompatible with Qwen2.5-VL. For prompt compression, we compare with FastV (Chen et al., 2024)—a method specialized for dropping image tokens inside LLMs. FastV uses a hyperparameter $k$: it runs all tokens up to layer $k$, then drops less-attended image tokens based on the attention map, processing only the top tokens thereafter. This makes FastV less efficient than SpecPC, since all tokens must be processed up to $k$ with the full model, requiring considerable memory. Notably, FastV must compute the entire attention map at layer $k$, preventing the use of FlashAttention and leading to out-of-memory errors, even for moderate sequence lengths. As a result, many MileBench datasets exceed 80GB VRAM, so we limit our analysis to these six datasets.

Since the selected MileBench datasets have relatively short average context lengths, we conduct experiments using reduced $C_{\max}$ values for both KV cache dropping (64, 96, and 128) and prompt compression (512, 768, and 1024).

Fig. 13a presents results for various KV dropping methods. Our proposed method, SpecKV, demonstrates performance comparable to existing approaches, while significantly outperforming the others on the WebQA task.

Fig. 13b compares the performance of SpecPC and FastV under two configurations ($k = 2$ and $k = 5$). Our method consistently outperforms FastV in most cases.

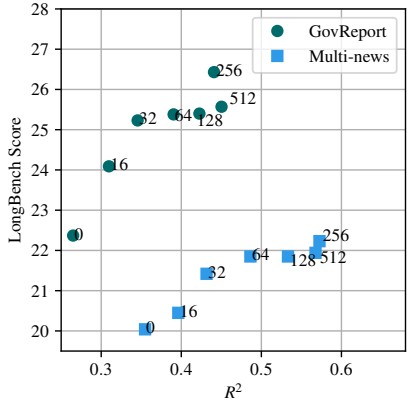

Figure 7: Impact of $n_{\text{lookahead}}$ on SpecKV importance score accuracy ($R^2$) and LongBench downstream performance ($n_{\text{out}} = 512$). The plot shows a strong positive correlation: as $n_{\text{lookahead}}$ increases (with $n_{\text{lookahead}} = 0$ being equivalent to SnapKV), both the $R^2$ (correlation with the true target model scores) and the downstream task score improve. Notably, SpecKV improves correlation and accuracy even with a small $n_{\text{lookahead}} = 32$, which is only 6.25% of the output length. Experiments use a Qwen2.5-0.5B draft model and a Qwen2.5-32B target model on two LongBench tasks.

### E.4 SPECKV: IMPACT OF $n_{\text{LOOKAHEAD}}$ PARAMETER ON IMPORTANCE SCORE CORRELATION

We analyze the impact of the $n_{\text{lookahead}}$ parameter on the correlation ($R^2$) between SpecKV's estimated importance scores and the ground-truth scores from the target model. For this analysis, we use 50 examples from the Multi-news and GovReport tasks from LongBench, selected for their long maximum output length (512 tokens). We employ Qwen2.5-0.5B as the draft model and Qwen2.5-32B as the target model, with $C_{\text{max}} = 256$.

To establish the ground-truth scores, we use the target model's generated output (up to $n_{\text{out}}$) as a perfect lookahead sequence and record the resulting importance scores. In Figs. 7, 14 and 15, we compute the $R^2$ correlation between these ground-truth scores and the scores estimated by SpecKV using various $n_{\text{lookahead}}$ values.

As shown in Figs. 14 and 15, the $R^2$ correlation steadily increases with $n_{\text{lookahead}}$, confirming that a larger lookahead provides a more accurate importance estimation. This benefit is particularly pronounced for longer output sequences ($n_{\text{out}}$). We also test $n_{\text{lookahead}} = 0$ (equivalent to SnapKV), which yields a significantly inferior correlation.

Furthermore, we connect this score accuracy to downstream performance. Fig. 7 plots the $R^2$ value against the final LongBench downstream score. We observe that these two metrics are highly correlated: a higher $R^2$ (better score accuracy) generally yields better downstream performance. Consequently, increasing $n_{\text{lookahead}}$ improves both the $R^2$ correlation and the final task score.

### E.5 PERFORMANCE OF CROSS-FAMILY MODELS

In this section, we evaluate the effectiveness of our methods using cross-family draft models. We set Llama-3.1-70B as the target model and employ Qwen2.5-0.5B and Qwen2.5B-1.5B as draft models. The evaluation is conducted on the RULER and LongBench benchmarks, using $C_{\text{max}} = 256$ for SpecKV and $C_{\text{max}} = 1024$ for SpecPC.

A key challenge in this setup is that the Qwen and Llama families use different tokenizers, necessitating a token translation step. For SpecKV, we de-tokenize the draft model's output and then re-tokenize it with the target model's tokenizer. For SpecPC, we aggregate attention scores from tokens to words before re-tokenizing, to avoid tokenizing partial words, and replace the draft model's chat template with the target's.

---

[21]https://github.com/bitsandbytes-foundation/bitsandbytes (MIT License)

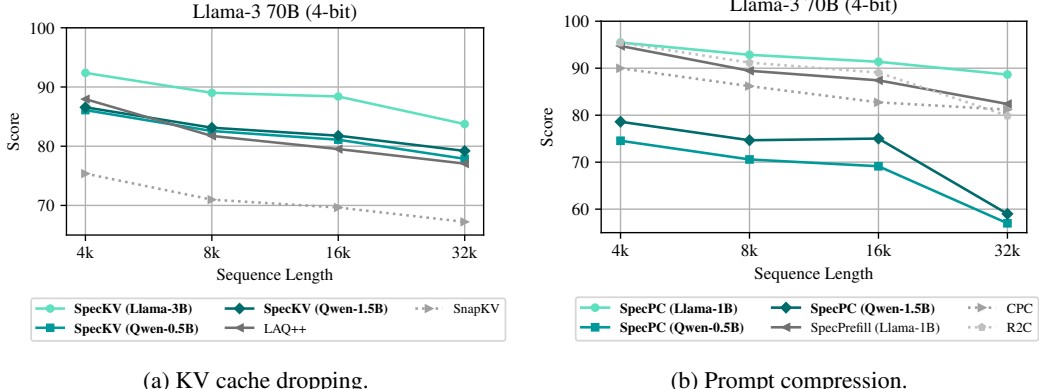

(a) KV cache dropping.

(b) Prompt compression.

Figure 8: Cross-family model results for SpecKV and SpecPC on RULER. SpecKV demonstrates robust performance even with draft models from different families, outperforming the strongest baseline, LAQ++.

Table 7: Cross-family model results for SpecKV and SpecPC on LongBench using Llama-3.1-70B (4-bit) as target model.

|  | $C_{\max}$ | Method | Single-doc QA | Multi-doc QA | Summary | Few-shot Learning | Code Completion | All |
|---|---|---|---|---|---|---|---|---|
| KV | 256 | SnapKV | **55.88** | 45.30 | 22.49 | 62.15 | 55.49 | 47.75 |
|  |  | LAQ++ | 54.90 | 46.48 | 22.83 | 64.31 | 55.10 | 48.43 |
|  |  | **SpecKV (Llama-3B)** | 51.80 | **47.23** | **25.53** | 64.02 | **58.75** | **48.80** |
|  |  | **SpecKV (Qwen-0.5B)** | 53.18 | 45.70 | 24.40 | 63.69 | 57.36 | 48.26 |
|  |  | **SpecKV (Qwen-1.5B)** | 51.20 | 47.07 | 24.98 | **65.81** | 57.53 | 48.73 |
| PC | 1024 | CPC | 45.14 | 39.41 | 24.86 | 61.40 | 37.58 | 41.97 |
|  |  | R2C | 48.93 | 42.01 | 25.38 | 58.91 | 40.19 | 43.29 |
|  |  | SpecPrefill (Llama-1B) | 54.62 | **46.43** | 25.63 | 64.80 | 44.92 | 48.37 |
|  |  | **SpecPC (Llama-1B)** | **56.84** | 44.48 | **25.91** | **67.37** | 47.15 | **48.44** |
|  |  | **SpecPC (Qwen-0.5B)** | 36.73 | 36.42 | 24.68 | 64.51 | **50.75** | 42.04 |
|  |  | **SpecPC (Qwen-1.5B)** | 51.58 | 39.53 | 25.87 | 66.43 | 50.25 | 46.48 |

The results, shown in Fig. 8 (RULER) and Table 7 (LongBench), indicate that both SpecKV and SpecPC achieve good results even with cross-family drafts. Notably, SpecKV outperforms the strongest baseline, LAQ++. This aligns with the intuition that SpecKV, which relies only on the draft model's output, should generalize well to cross-model scenarios. SpecPC experiences a performance drop because it relies on similar attention patterns between the draft and target models, a condition that may not hold in cross-family setups.

### E.6 Extended Results for SpecKV-PC

This section presents extended results for the cascaded compression strategy, SpecKV-PC. By restricting the large target model to process only a small fraction of the original prompt, this approach achieves substantially better latency and memory efficiency than SpecKV alone.

We evaluate the impact of varying prompt compression ratios on accuracy using Qwen2.5-1.5B/32B and Llama-3-3B/70B as draft/target pairs, with a final KV cache size of $C_{\max} = 256$. Our combined method, denoted as SpecKV-PC-X (where the prompt is compressed to X tokens), is benchmarked against standard SpecKV, SpecPC, LAQ++, and SnapKV.

As illustrated in Fig. 9 (RULER) and Table 8 (LongBench), SpecKV-PC proves to be both efficient and highly accurate. Notably, we find that moderate pre-compression (e.g., SpecKV-PC-2048) yields accuracy superior to standard SpecKV alone, particularly at the longer sequence lengths (RULER). This suggests that the initial SpecPC stage acts as an effective pre-filter, discarding irrelevant information to help the target model focus on the most important tokens.

Conversely, extreme prompt compression (e.g., SpecPC-256) results in significant performance degradation. This indicates that for aggressive compression targets, coarse prompt-level reduction is insufficient.

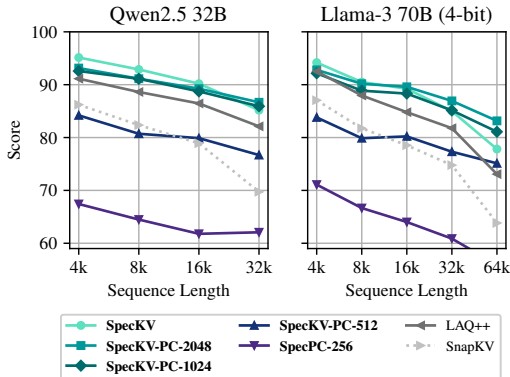

Figure 9: Performance of the combined SpecKV and SpecPC methods on RULER (final $C_{max} = 256$). Our cascaded approach (SpecKV-PC-X) pre-compresses the prompt to X tokens, achieving higher accuracy than standard SpecKV (which sees the full prompt) at longer sequences. This suggests the initial pre-filtering allows SpecKV to make a more effective final selection.

Table 8: LongBench performance of cascaded compression with SpecKV and SpecPC using Qwen2.5 and Llama-3. SpecKV-PC-2048 achieves superior performance.

| | | Qwen2.5 32B | | | | | | Llama-3 70B (4-bit) | | | | | |
| Group | Method | SingleQA | MultiQA | Summ. | Few-shot | Code | All | SingleQA | MultiQA | Summ. | Few-shot | Code | All |
|---|---|---|---|---|---|---|---|---|---|---|---|---|---|
| Dense | Target | 56.01 | 43.99 | 25.90 | 64.06 | 44.74 | 47.78 | 55.02 | 47.06 | 28.61 | 70.47 | 48.19 | 49.99 |
| KV | SnapKV | 52.54 | 40.21 | 19.89 | 61.18 | 40.12 | 42.98 | 55.88 | 45.30 | 22.49 | 62.15 | 55.49 | 47.75 |
| | LAQ++ | 55.15 | 44.14 | 22.24 | 63.25 | 41.19 | 45.79 | 54.90 | 46.48 | 22.83 | 64.31 | 55.10 | 48.43 |
| | **SpecKV** | 53.48 | 43.77 | 24.02 | **63.79** | **44.80** | **46.06** | 51.80 | 47.23 | 25.53 | 64.02 | **58.75** | 48.80 |
| | **SpecKV-PC-2048** | 52.60 | **44.52** | 24.11 | 63.38 | 48.45 | **46.48** | **61.42** | 47.15 | **26.51** | **66.94** | 58.19 | **51.60** |
| | **SpecKV-PC-1024** | **55.37** | 42.21 | **24.14** | 63.09 | 39.72 | 45.28 | 58.73 | **47.33** | 25.69 | 64.28 | 55.19 | 49.89 |
| | **SpecKV-PC-512** | 45.77 | 36.57 | 22.36 | 56.57 | 39.57 | 40.21 | 51.73 | 46.11 | 23.01 | 63.64 | 50.00 | 46.68 |
| PC | **SpecPC-256** | 34.75 | 27.05 | 19.15 | 45.0 | 35.07 | 32.0 | 27.97 | 25.93 | 19.09 | 51.25 | 48.12 | 33.5 |

## E.7 SPECKV: EXTENDED LATENCY AND MEMORY ANALYSIS

We analyze the latency and memory usage of baselines and our algorithms, breaking down latency into three stages: draft generation, target prefill, and target decoding. For LAQ++, draft generation occurs after target prefill to share the same KV cache.

All experiments are run on an H200 GPU (141 GB VRAM) using the Qwen2.5 model (3B draft, 32B target). We evaluate across a range of input sizes, $n_{in} \in \{4k, 8k, 16k, 32k, 64k\}$, and output sizes, $n_{out} \in \{64, 128, 256, 512\}$. A global $C_{max} = 256$ is used for all experiments. For algorithm-specific settings, we set $n_{lookahead} = n_{out}$ for SpecKV and use the default $n_{lookahead} = 8$ for LAQ++. For the SpecKV-PC variant, we employ SpecKV-PC-2048, which pre-compresses the prompt to 2048 tokens.

Our results demonstrate that draft lookahead introduces negligible latency (Fig. 16) and memory overhead (Fig. 17) relative to the total cost of target model inference. Notably, cascaded compression with SpecKV and SpecPC (SpecKV-PC) significantly reduces prefill time and peak memory. At 64k context, SpecKV-PC is about 40% faster and 25 GB more memory efficient than LAQ++.

## E.8 ABLATION STUDIES

In this section, we conduct a series of ablation experiments to further analyze the effectiveness of our two proposed methods: SpecKV and SpecPC. For consistency, we fix $C_{max}$ to 256 for KV dropping

and 1024 for prompt compression across all experiments. We utilize Qwen2.5 (Instruct), employing the 0.5B model as the draft and the 14B model as the target. We sample 100 random examples per task from the LongBench and RULER benchmarks.

### E.8.1 SPECKV: ANALYSIS OF ENHANCED DRAFT MODELS

Fig. 18 illustrates the impact of using draft models of different sizes and versions on the RULER score. As anticipated, both newer (Yang et al., 2025a) and larger draft models lead to improved performance of SpecKV.

### E.8.2 SPECKV AND SPECPC: IMPACT OF $n_{\text{LOOKAHEAD}}$

Fig. 19 illustrates how varying the number of generated draft tokens, $n_{\text{lookahead}}$, affects the performance of SpecKV and SpecPC. Overall, increasing $n_{\text{lookahead}}$ generally results in higher final accuracy for SpecKV, whereas it yields only marginal improvements for SpecPC. We attribute this to the larger $C_{\text{max}}$ budget of SpecPC, which allows it to capture all important tokens without needing to generate long drafts.

### E.8.3 SPECKV: ACCURACY ANALYSIS OF SPARSE PREFILL

In this section, we experimentally evaluate how the sparsity of SpecKV's prefill procedure affects downstream task performance (Fig. 20). Specifically, we set $n_{\text{vert}}$ equal to $n_{\text{slash}}$ and compare several values for these parameters. As anticipated, reducing sparsity (i.e., using a higher $n_{\text{vert}}$) generally results in higher accuracy; however, accuracy improvements plateau at $n_{\text{vert}} = 2048$, which we therefore adopt for our main experiments.

Interestingly, for certain LongBench categories, increased sparsity (i.e., lower $n_{\text{vert}}$) can actually lead to improved performance. This counterintuitive result suggests that, for some tasks, sparser prefill may serve as a form of regularization, preventing overfitting to irrelevant context.

## LLM USAGE DISCLOSURE

We utilized Google's Gemini 2.5 Pro (Google DeepMind, 2025) and OpenAI's GPT (Achiam et al., 2023) to assist with improving the grammar, clarity, and readability of this manuscript. The authors reviewed and edited all LLM-generated suggestions to ensure the final text accurately reflects our scientific contributions and claims. The authors retain full responsibility for the content of this paper.

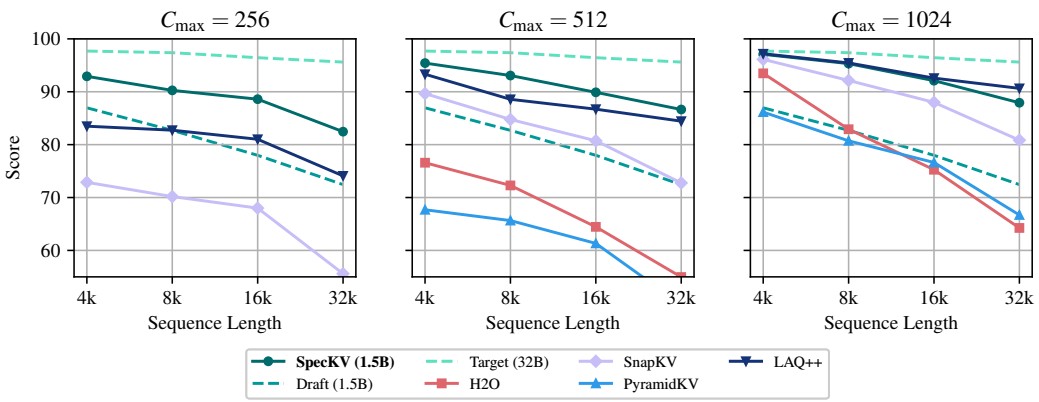

(a) Qwen2.5: Draft model (1.5B), target model (32B)[22]

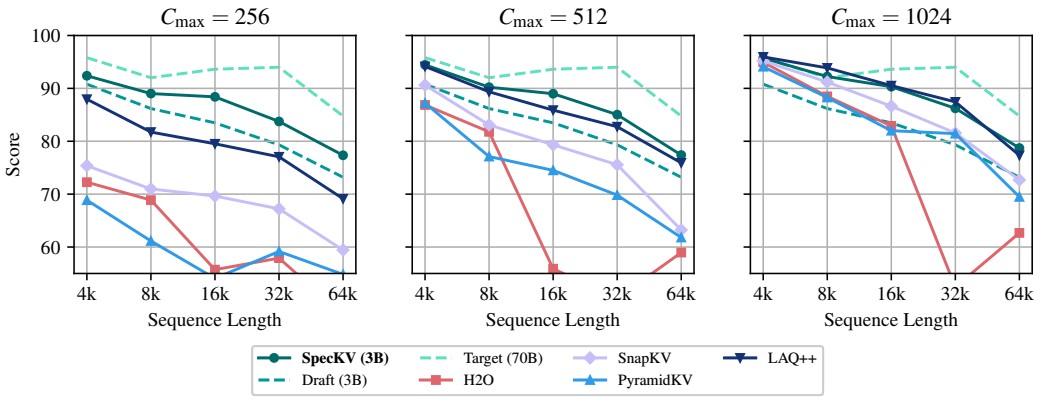

(b) Llama-3: Draft model (3.2-3B), target model (3.1-70B, 4-bit)

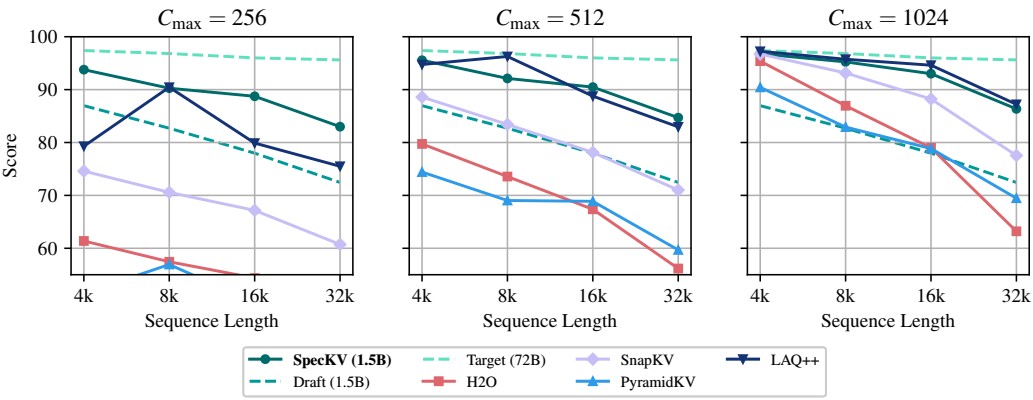

(c) Qwen2.5: Draft model (1.5B), target model (72B, 4-bit)

Figure 10: Extended RULER results for KV cache dropping. Our proposed SpecKV method consistently outperforms SnapKV and LAQ++ across the majority of evaluated settings, often by a substantial margin.

---

[22]H2O and PyramidKV are not plotted for Qwen2.5 at $C_{max} = 256$ as their performance falls outside the visible range.

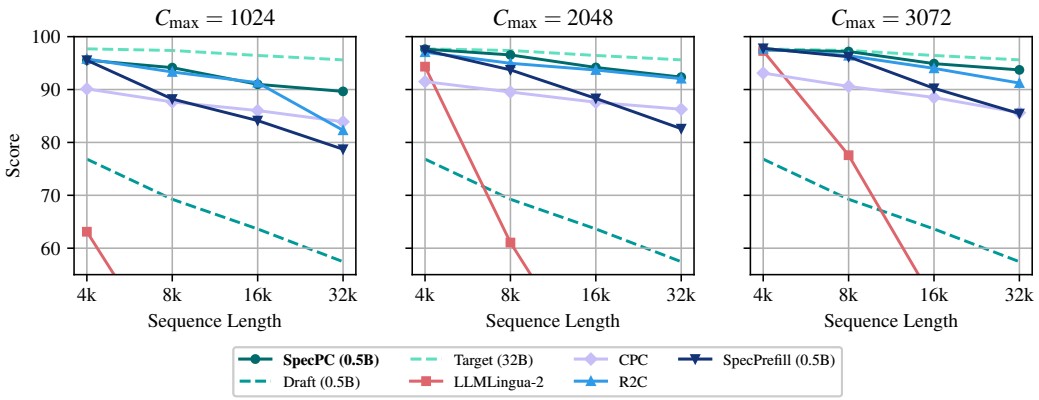

(a) Qwen2.5: Draft model (0.5B), target model (32B)

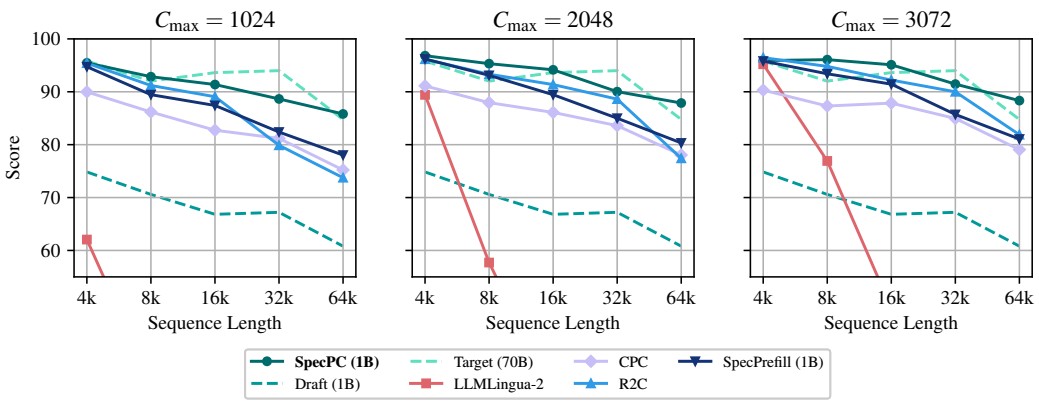

(b) Llama-3: Draft model (3.2-1B), target model (3.1-70B, 4-bit)

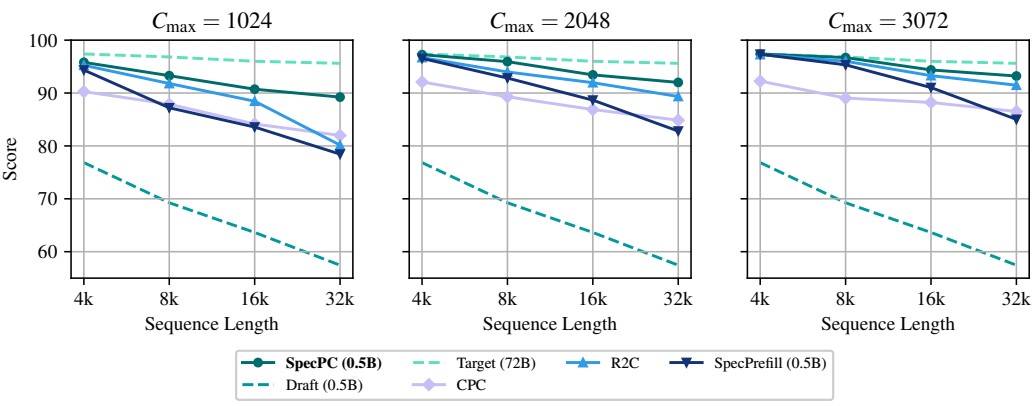

(c) Qwen2.5: Draft model (0.5B), target model (72B, 4-bit)

Figure 11: Extended RULER results on prompt compression. Our proposed SpecPC method consistently outperforms CPC, R2C, and SpecPrefill, maintaining strong performance even on long sequences.

Table 9: LongBench results for Qwen2.5, featuring 0.5B (SpecPC) and 1.5B (SpecKV) draft models and a 32B target model.

| | $C_{\max}$ | Method | Single-doc QA | Multi-doc QA | Summary | Few-shot Learning | Code Completion | All |
|---|---|---|---|---|---|---|---|---|
| Dense | – | Draft (0.5B) | 19.07 | 26.58 | 20.90 | 53.51 | 32.48 | 30.37 |
| | | Draft (1.5B) | 36.16 | 35.01 | 22.79 | 63.92 | 36.62 | 39.06 |
| | | Target (32B) | 56.01 | 43.99 | 25.90 | 64.06 | 44.74 | 47.78 |
| KV | 256 | H2O | 46.63 | 30.81 | 19.88 | 56.03 | 39.27 | 39.32 |
| | | SnapKV | 52.54 | 40.21 | 19.89 | 61.18 | 40.12 | 42.98 |
| | | PyramidKV | 50.92 | 37.26 | 18.90 | 63.24 | 40.20 | 43.19 |
| | | LAQ++ | **55.15** | **44.14** | 22.24 | 63.25 | 41.19 | 45.79 |
| | | **SpecKV (1.5B)** | 53.48 | 43.77 | **24.02** | **63.79** | **44.80** | **46.06** |
| | 512 | H2O | 48.62 | 36.72 | 21.31 | 59.10 | 40.99 | 42.27 |
| | | SnapKV | 55.24 | 42.21 | 21.47 | 63.36 | 39.69 | 44.73 |
| | | PyramidKV | 54.79 | 41.87 | 20.28 | 62.00 | 40.08 | 46.55 |
| | | LAQ++ | **55.77** | **45.30** | 23.96 | 62.37 | 41.86 | 46.44 |
| | | **SpecKV (1.5B)** | 52.78 | 43.97 | **24.80** | **64.72** | **46.77** | **46.60** |
| | 1024 | H2O | 51.86 | 41.30 | 23.02 | 60.18 | 42.93 | 44.84 |
| | | SnapKV | 55.32 | 44.04 | 23.08 | **66.15** | **44.42** | 46.76 |
| | | PyramidKV | 55.91 | 42.67 | 22.28 | 64.76 | 42.52 | **48.31** |
| | | LAQ++ | **56.33** | **45.18** | 25.17 | 62.13 | 43.06 | 46.61 |
| | | **SpecKV (1.5B)** | 55.72 | 43.95 | **25.20** | 63.31 | 42.38 | 46.38 |
| PC | 1024 | CPC | 45.60 | 40.62 | 23.09 | 60.08 | 32.31 | 40.91 |
| | | R2C | 50.49 | 40.37 | 23.26 | 53.45 | 34.11 | 39.88 |
| | | SpecPrefill (0.5B) | 45.94 | 39.32 | 23.16 | 62.04 | **43.17** | 42.70 |
| | | **SpecPC (0.5B)** | **51.23** | **41.40** | **23.37** | **62.26** | 38.23 | **43.66** |
| | 2048 | CPC | 51.01 | 42.31 | 23.74 | 60.92 | 35.83 | 43.26 |
| | | R2C | 50.32 | **42.66** | 24.08 | 59.11 | 40.54 | 44.19 |
| | | SpecPrefill (0.5B) | 51.60 | 40.72 | 23.71 | **64.20** | 45.29 | 45.09 |
| | | **SpecPC (0.5B)** | **55.40** | 42.60 | **24.12** | 61.46 | **48.05** | **46.20** |
| | 3072 | CPC | 56.80 | 42.05 | **24.77** | 62.79 | 37.98 | 45.37 |
| | | R2C | 51.67 | 42.88 | 24.09 | 62.45 | 27.01 | 42.66 |
| | | SpecPrefill (0.5B) | 53.85 | **42.92** | 24.48 | **64.03** | 45.76 | 46.24 |
| | | **SpecPC (0.5B)** | **56.89** | 42.42 | 24.71 | 62.66 | **47.49** | **46.79** |

Table 10: LongBench results for Llama-3, featuring 3.2-1B (SpecPC) and 3.2-3B (SpecKV) draft models and a 3.1-70B (4-bit) target model.

| | $C_{\max}$ | Method | Single-doc QA | Multi-doc QA | Summary | Few-shot Learning | Code Completion | All |
|---|---|---|---|---|---|---|---|---|
| Dense | – | Draft (1B) | 28.37 | 28.62 | 26.12 | 59.10 | 33.59 | 35.27 |
| | | Draft (3B) | 45.53 | 40.52 | 27.46 | 64.82 | 46.73 | 44.89 |
| | | Target (70B) | 55.02 | 47.06 | 28.61 | 70.47 | 48.19 | 49.99 |
| KV | 256 | H2O | 54.07 | 41.30 | 22.55 | 49.10 | 54.14 | 43.52 |
| | | SnapKV | **55.88** | 45.30 | 22.49 | 62.15 | 55.49 | 47.75 |
| | | PyramidKV | 55.41 | 45.59 | 22.50 | 59.06 | 49.90 | 46.25 |
| | | LAQ++ | 54.90 | 46.48 | 22.83 | **64.31** | 55.10 | 48.43 |
| | | **SpecKV (3B)** | 51.80 | **47.23** | **25.53** | 64.02 | **58.75** | **48.80** |
| | 512 | H2O | 55.01 | 43.86 | 23.85 | 58.48 | 52.00 | 46.26 |
| | | SnapKV | **56.08** | 46.94 | 24.40 | 63.34 | 52.14 | 48.32 |
| | | PyramidKV | 54.73 | 47.07 | 24.20 | 63.62 | 47.05 | 47.35 |
| | | LAQ++ | 55.06 | 47.45 | 24.58 | 63.39 | 51.75 | 48.36 |
| | | **SpecKV (3B)** | 53.31 | **47.47** | **27.33** | **67.49** | **54.19** | **49.66** |
| | 1024 | H2O | 54.19 | 45.77 | 26.03 | 63.62 | 49.58 | 47.71 |
| | | SnapKV | 54.94 | **47.40** | 25.97 | **67.56** | 48.13 | **48.85** |
| | | PyramidKV | **56.32** | 46.82 | 26.20 | 62.58 | 48.28 | 48.02 |
| | | LAQ++ | 55.19 | 47.17 | 26.43 | 67.41 | 47.16 | 48.61 |
| | | **SpecKV (3B)** | 53.32 | 46.02 | **27.41** | 66.40 | **50.66** | 48.63 |
| PC | 1024 | CPC | 45.14 | 39.41 | 24.86 | 61.40 | 37.58 | 41.97 |
| | | R2C | 48.93 | 42.01 | 25.38 | 58.91 | 40.19 | 43.29 |
| | | SpecPrefill (1B) | 54.62 | **46.43** | 25.63 | 64.80 | 44.92 | 48.37 |
| | | **SpecPC (1B)** | **56.84** | 44.48 | **25.91** | **67.37** | **47.15** | **48.44** |
| | 2048 | CPC | 55.97 | 46.00 | 26.72 | 64.78 | 41.31 | 47.36 |
| | | R2C | 53.62 | **46.70** | 26.27 | 62.41 | **47.90** | 47.34 |
| | | SpecPrefill (1B) | 58.39 | 45.37 | 27.13 | 66.60 | 45.45 | 48.81 |
| | | **SpecPC (1B)** | **59.39** | 46.25 | **27.60** | **68.42** | 46.70 | **49.88** |
| | 3072 | CPC | 56.64 | 46.69 | 27.29 | 64.92 | 44.75 | 48.30 |
| | | R2C | 56.11 | 45.15 | 27.51 | 61.76 | **47.17** | 47.57 |
| | | SpecPrefill (1B) | 57.75 | 46.15 | 27.15 | 64.70 | 42.50 | 48.02 |
| | | **SpecPC (1B)** | **58.47** | **48.07** | **27.72** | 65.43 | 41.28 | **48.69** |

Table 11: LongBench results for Qwen2.5, featuring 0.5B (SpecPC) and 1.5B (SpecKV) draft models and a 72B (4-bit) target model.

| | $C_{\max}$ | Method | Single-doc QA | Multi-doc QA | Summary | Few-shot Learning | Code Completion | All |
|---|---|---|---|---|---|---|---|---|
| Dense | – | Draft (0.5B) | 19.07 | 26.58 | 20.90 | 53.51 | 32.48 | 30.37 |
| | | Draft (1.5B) | 36.16 | 35.01 | 22.79 | 63.92 | 36.62 | 39.06 |
| | | Target (72B) | 58.70 | 45.89 | 26.24 | 64.83 | 51.08 | 49.22 |
| KV | 256 | H2O | 53.68 | 35.05 | 21.10 | 50.84 | 48.87 | 41.41 |
| | | SnapKV | 55.85 | 40.12 | 21.65 | 55.82 | 50.41 | 44.37 |
| | | PyramidKV | 55.13 | 38.89 | 20.14 | 53.02 | 49.59 | 42.91 |
| | | LAQ++ | **57.26** | **43.54** | 22.95 | 62.53 | **55.79** | 46.98 |
| | | **SpecKV (1.5B)** | 53.94 | 42.71 | **25.07** | **66.58** | 46.93 | **47.06** |
| | 512 | H2O | 55.77 | 39.52 | 22.61 | 58.01 | 50.69 | 44.94 |
| | | SnapKV | **59.08** | **44.13** | 23.04 | 61.33 | 51.77 | 47.59 |
| | | PyramidKV | 56.98 | 42.52 | 22.03 | 55.43 | 50.29 | 45.10 |
| | | LAQ++ | 58.28 | 34.94 | 24.38 | 63.74 | **54.79** | **49.08** |
| | | **SpecKV (1.5B)** | 55.74 | 43.39 | **25.85** | 64.56 | 47.80 | 47.44 |
| | 1024 | H2O | 55.37 | 43.50 | 24.11 | 62.09 | 50.68 | 46.90 |
| | | SnapKV | **59.56** | 44.06 | 24.41 | 62.85 | 52.87 | 48.46 |
| | | PyramidKV | 58.34 | 43.68 | 23.18 | 59.66 | 51.45 | 46.96 |
| | | LAQ++ | 58.66 | 35.37 | **26.35** | 60.90 | **53.56** | **49.59** |
| | | **SpecKV (1.5B)** | 56.04 | **44.27** | 25.71 | 63.46 | 49.90 | 47.73 |
| PC | 1024 | CPC | 46.37 | 38.36 | 24.19 | 52.54 | 28.30 | 38.64 |
| | | R2C | **55.78** | 40.10 | 24.40 | 47.62 | 33.17 | 40.72 |
| | | SpecPrefill (0.5B) | 49.04 | 40.32 | **24.41** | **63.53** | **50.12** | 45.16 |
| | | **SpecPC (0.5B)** | 48.52 | **45.34** | 24.14 | 61.04 | 48.30 | **45.26** |
| | 2048 | CPC | 55.14 | 43.53 | 24.71 | 49.46 | 33.21 | 41.78 |
| | | R2C | 54.61 | 45.70 | 24.99 | 53.65 | 42.43 | 44.41 |
| | | SpecPrefill (0.5B) | 53.39 | 43.46 | **25.53** | **65.89** | 50.18 | 47.51 |
| | | **SpecPC (0.5B)** | **58.36** | **46.38** | 25.32 | 64.43 | **51.11** | **48.98** |
| | 3072 | CPC | 55.78 | 44.41 | 25.47 | 42.71 | 39.16 | 41.68 |
| | | R2C | 59.22 | 44.70 | 25.69 | 55.22 | 44.04 | 45.90 |
| | | SpecPrefill (0.5B) | 55.75 | **45.71** | **25.93** | **64.22** | **51.63** | 48.44 |
| | | **SpecPC (0.5B)** | **59.84** | 44.95 | 25.91 | 64.18 | 46.88 | **48.46** |

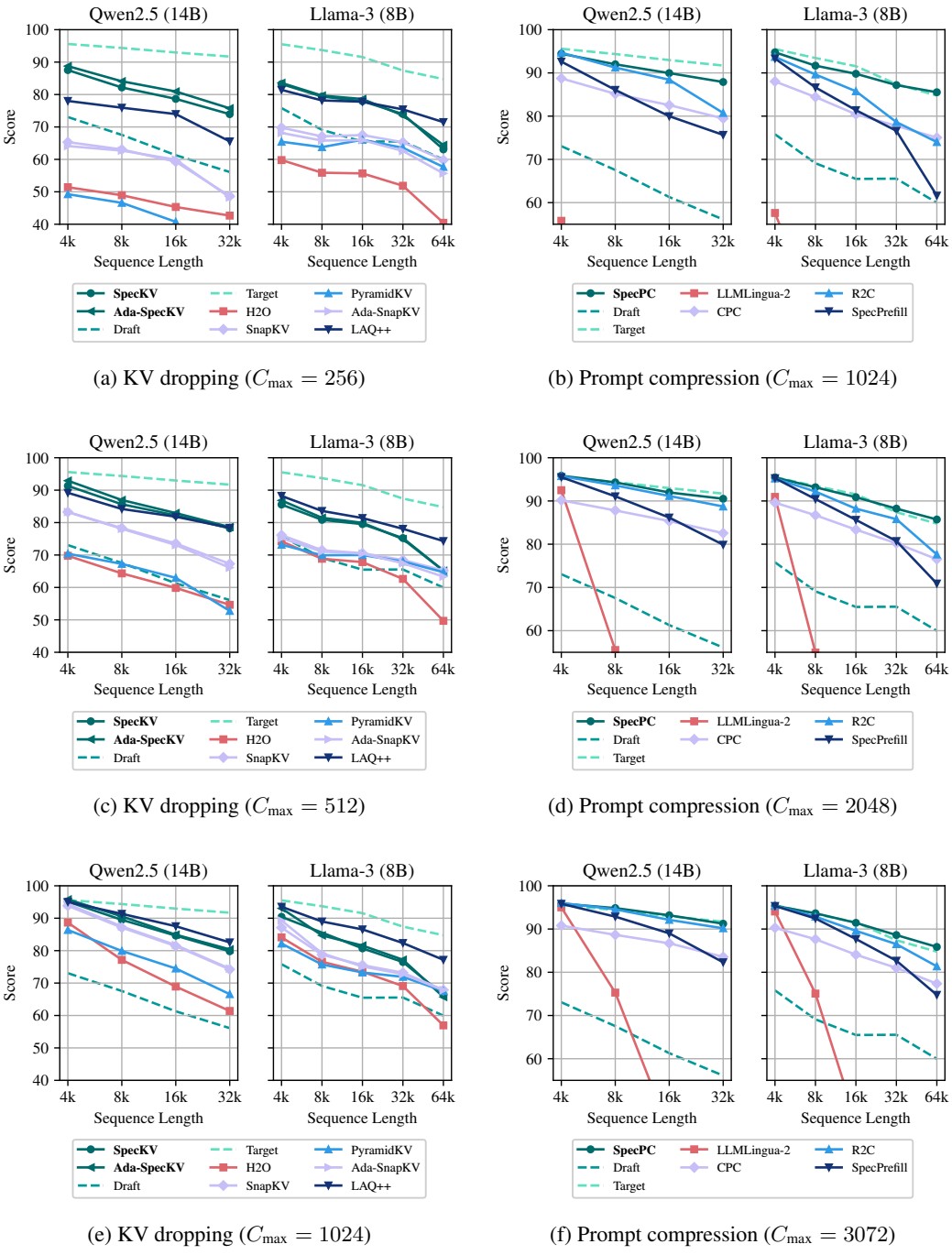

Figure 12: Performance of the proposed SpecKV and SpecPC on RULER compared to various baselines including AdaKV for Qwen2.5-0.5B-Instruct and Llama-3.2-1B-Instruct as draft models, and Qwen2.5-14B-Instruct and Llama-3.1-8B-Instruct as target models. AdaKV can further boost the performance of SpecKV. Notably, SpecPC performs nearly on par with the target model. Low-performing methods are omitted.

Table 12: Results for LongBench performance with Qwen2.5-0.5B (draft) and Qwen2.5-14B (target).

| | $C_{\max}$ | Method | Single-doc QA | Multi-doc QA | Summary | Few-shot Learning | Code Completion | All |
|---|---|---|---|---|---|---|---|---|
| Dense | – | Draft | 21.04 | 24.4 | 21.23 | 54.07 | 33.39 | 30.64 |
| | | Target | 53.19 | 42.83 | 24.99 | 65.34 | 51.75 | 47.33 |
| KV | 256 | StreamingLLM | 38.66 | 24.13 | 19.10 | 49.75 | 35.23 | 33.24 |
| | | H2O | 46.98 | 29.66 | 19.82 | 50.88 | 47.86 | 38.41 |
| | | SnapKV | 49.07 | 33.19 | 19.49 | 54.49 | 47.34 | 40.25 |
| | | PyramidKV | 47.07 | 31.84 | 18.32 | 54.81 | 43.26 | 38.76 |
| | | Ada-SnapKV | 50.92 | 34.57 | 19.63 | 55.07 | 49.58 | 41.41 |
| | | LAQ++ | **52.45** | **39.99** | 22.12 | **61.46** | 51.32 | 45.05 |
| | | **SpecKV** | 51.07 | 38.76 | 23.20 | 59.68 | 49.58 | 44.09 |
| | | **Ada-SpecKV** | 51.92 | 39.38 | **24.90** | 61.05 | **53.43** | **45.61** |
| | 512 | StreamingLLM | 38.97 | 27.04 | 21.14 | 53.78 | 36.52 | 35.41 |
| | | H2O | 47.56 | 33.16 | 21.02 | 53.45 | 49.78 | 40.37 |
| | | SnapKV | 50.74 | 38.33 | 20.97 | 57.81 | 49.94 | 43.10 |
| | | PyramidKV | 50.19 | 36.37 | 20.25 | 58.17 | 47.88 | 42.19 |
| | | Ada-SnapKV | 51.69 | 38.61 | 21.48 | 59.84 | 51.85 | 44.18 |
| | | LAQ++ | **52.31** | 42.06 | 23.75 | 61.62 | 51.64 | 45.89 |
| | | **SpecKV** | 51.98 | 41.51 | 24.11 | 62.60 | 52.33 | 46.09 |
| | | **Ada-SpecKV** | 51.00 | **42.18** | **25.74** | **63.74** | **55.28** | **48.46** |
| | 1024 | StreamingLLM | 40.08 | 32.01 | 22.12 | 55.42 | 38.36 | 37.54 |
| | | H2O | 49.73 | 37.27 | 22.38 | 55.94 | 51.26 | 42.75 |
| | | SnapKV | 51.56 | 40.98 | 22.65 | 63.90 | 51.48 | 45.73 |
| | | PyramidKV | 51.42 | 40.93 | 21.76 | 59.96 | 49.92 | 44.43 |
| | | Ada-SnapKV | 51.81 | 40.96 | 22.83 | 64.14 | 51.97 | 45.94 |
| | | LAQ++ | 52.27 | 42.59 | 25.04 | 61.56 | 51.68 | 46.27 |
| | | **SpecKV** | **52.92** | 41.85 | 24.30 | 63.46 | 52.50 | 46.61 |
| | | **Ada-SpecKV** | 52.27 | **43.22** | **26.39** | **64.83** | **54.40** | **47.78** |
| PC | 1024 | LLMLingua-2 | 27.10 | 23.74 | 22.57 | 35.85 | 37.62 | 28.79 |
| | | CPC | 42.76 | 36.05 | 22.58 | 48.38 | 35.57 | 37.17 |
| | | R2C | 46.43 | 37.42 | 22.64 | 47.21 | 29.48 | 37.15 |
| | | SpecPrefill | 46.21 | 37.21 | 22.82 | **62.18** | 48.34 | 43.00 |
| | | **SpecPC** | **47.70** | **38.30** | **23.30** | 59.74 | **52.52** | **43.73** |
| | 2048 | LLMLingua-2 | 35.10 | 32.22 | 23.51 | 39.82 | 44.80 | 34.40 |
| | | CPC | 48.74 | 39.70 | 23.39 | 55.77 | 42.07 | 41.92 |
| | | R2C | 50.68 | 40.65 | 23.47 | 56.72 | 45.62 | 43.27 |
| | | SpecPrefill | 49.75 | 38.58 | **23.70** | **65.05** | 50.40 | 45.15 |
| | | **SpecPC** | **53.23** | **41.43** | 23.68 | 63.26 | **54.92** | **46.76** |
| | 3072 | LLMLingua-2 | 43.50 | 35.67 | 24.10 | 45.66 | 47.32 | 38.67 |
| | | CPC | 51.59 | 41.08 | 23.84 | 58.14 | 44.82 | 43.83 |
| | | R2C | 51.71 | 41.39 | 23.89 | 62.09 | 48.58 | 45.31 |
| | | SpecPrefill | 51.81 | 41.27 | 24.26 | 64.28 | 50.66 | 46.15 |
| | | **SpecPC** | **53.25** | **41.48** | **24.49** | **64.51** | **54.41** | **47.14** |

Table 13: Results for LongBench performance with Llama-3.2-1B (draft) and Llama-3.1-8B (target).

| | $C_{\max}$ | Method | Single-doc QA | Multi-doc QA | Summary | Few-shot Learning | Code Completion | All |
|---|---|---|---|---|---|---|---|---|
| Dense | – | Draft | 28.08 | 27.27 | 25.65 | 60.16 | 31.11 | 34.69 |
| | | Target | 45.85 | 43.79 | 28.68 | 66.65 | 50.46 | 46.84 |
| KV | 256 | StreamingLLM | 38.69 | 27.12 | 21.64 | 50.75 | 34.74 | 34.58 |
| | | H2O | 43.54 | 36.81 | 22.62 | 55.64 | 47.81 | 40.82 |
| | | SnapKV | 43.79 | 37.31 | 21.96 | 56.29 | 47.34 | 40.91 |
| | | PyramidKV | 43.62 | 37.79 | 21.75 | 55.15 | 46.32 | 40.54 |
| | | Ada-SnapKV | 44.04 | 37.86 | 22.34 | 59.95 | 50.39 | 42.38 |
| | | LAQ++ | **45.81** | **41.72** | 22.93 | **64.18** | 47.21 | **44.17** |
| | | **SpecKV** | 43.23 | 39.73 | 24.43 | 60.90 | 51.09 | 43.36 |
| | | **Ada-SpecKV** | 42.40 | 40.73 | **25.74** | 57.94 | **52.51** | 43.25 |
| | 512 | StreamingLLM | 38.80 | 29.15 | 24.16 | 52.59 | 37.25 | 36.33 |
| | | H2O | 44.82 | 39.36 | 23.95 | 58.70 | 49.30 | 42.79 |
| | | SnapKV | 45.00 | 41.31 | 23.61 | 61.36 | 49.92 | 43.83 |
| | | PyramidKV | 45.26 | 41.22 | 23.36 | 61.15 | 48.06 | 43.51 |
| | | Ada-SnapKV | 45.09 | 41.17 | 24.19 | 61.81 | 51.74 | 44.30 |
| | | LAQ++ | **46.17** | **43.44** | 24.17 | **63.13** | 48.74 | **44.87** |
| | | **SpecKV** | 43.48 | 41.86 | 26.17 | 62.34 | 51.36 | 44.59 |
| | | **Ada-SpecKV** | 43.80 | 42.56 | **26.59** | 58.72 | **54.41** | 44.56 |
| | 1024 | StreamingLLM | 38.45 | 32.54 | 25.03 | 58.09 | 38.60 | 38.54 |
| | | H2O | 45.45 | 42.50 | 25.42 | 59.17 | 50.10 | 44.13 |
| | | SnapKV | 45.46 | 43.15 | 25.42 | 62.06 | 52.37 | 45.22 |
| | | PyramidKV | 46.10 | 42.78 | 25.17 | 63.43 | 49.55 | 45.11 |
| | | Ada-SnapKV | 46.06 | 43.34 | 25.48 | 63.47 | 50.50 | 45.43 |
| | | LAQ++ | **46.21** | **44.14** | 25.71 | **64.10** | 50.00 | **45.75** |
| | | **SpecKV** | 43.73 | 43.39 | 26.95 | 63.14 | 51.28 | 45.30 |
| | | **Ada-SpecKV** | 44.83 | 43.72 | **27.50** | 61.04 | **54.23** | 45.70 |
| PC | 1024 | LLMLingua-2 | 29.61 | 24.83 | 23.43 | 24.18 | 40.66 | 27.68 |
| | | CPC | 35.67 | 36.61 | 25.26 | 34.01 | 43.58 | 34.42 |
| | | R2C | 38.41 | 39.07 | 25.28 | 43.26 | 43.99 | 37.58 |
| | | SpecPrefill | 41.51 | 39.38 | 25.82 | 63.51 | 44.68 | 42.86 |
| | | **SpecPC** | **44.83** | **39.94** | **25.85** | 63.70 | **44.82** | **43.76** |
| | 2048 | LLMLingua-2 | 34.00 | 32.51 | 24.90 | 24.76 | **47.27** | 31.64 |
| | | CPC | 40.02 | 39.41 | 26.83 | 39.02 | 46.66 | 37.80 |
| | | R2C | 44.53 | 38.97 | 26.63 | 54.62 | 46.67 | 41.97 |
| | | SpecPrefill | 42.35 | **42.11** | 27.23 | 63.56 | 44.46 | 43.91 |
| | | **SpecPC** | **44.92** | 40.71 | **27.30** | 64.77 | 46.89 | **44.78** |
| | 3072 | LLMLingua-2 | 39.13 | 35.44 | 25.98 | 29.73 | **49.92** | 35.05 |
| | | CPC | 41.73 | 39.52 | 27.27 | 42.12 | 49.13 | 39.30 |
| | | R2C | 44.77 | 40.97 | 27.35 | 60.85 | 48.08 | 44.14 |
| | | SpecPrefill | 43.86 | **42.45** | 27.82 | 63.17 | 45.25 | 44.46 |
| | | **SpecPC** | **47.12** | 41.95 | **28.02** | 65.61 | 45.52 | **45.65** |

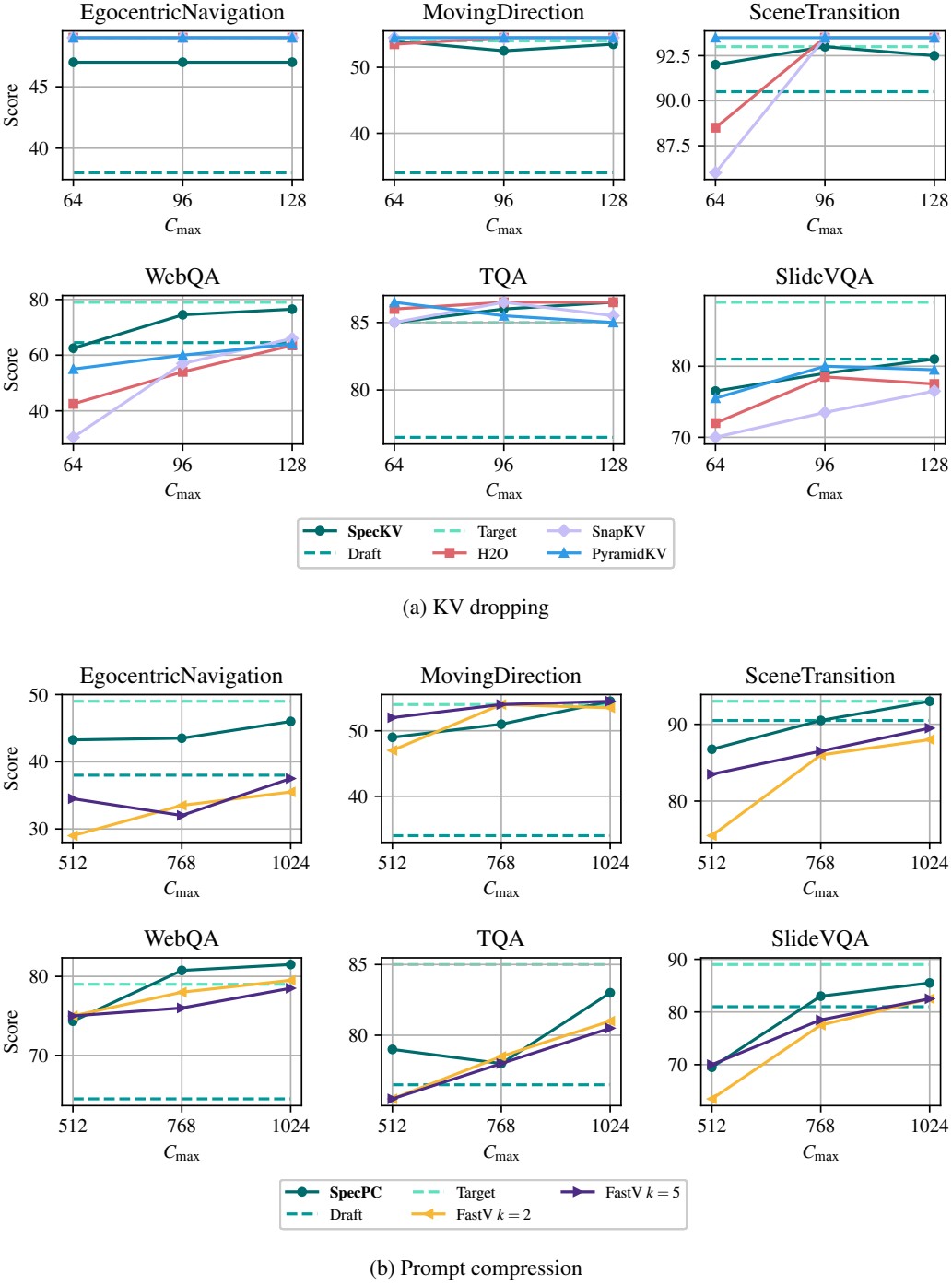

(a) KV dropping

(b) Prompt compression

Figure 13: MileBench multi-modal results using Qwen2.5-VL-3B-Instruct-AWQ (draft) and Qwen2.5-VL-32B-Instruct-AWQ (target). SpecKV demonstrates competitive performance across most tasks and achieves a substantial improvement on WebQA. SpecPC consistently outperforms both FastV configurations on the majority of datasets.

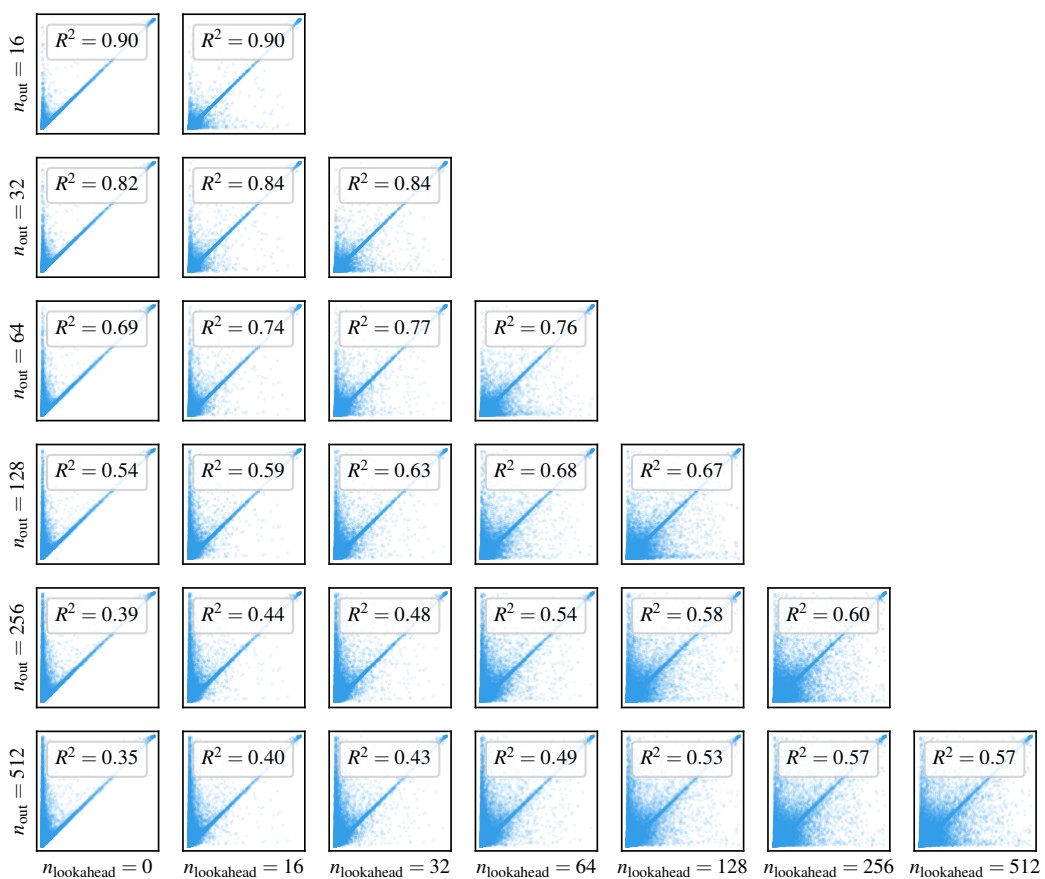

Figure 14: Correlation of importance scores from SpecKV against ground-truth scores from the dense target model for various output lengths ($n_{out}$) and lookaheads ($n_{lookahead}$). Increasing $n_{lookahead}$ improves correlation, especially for longer output lengths ($n_{out}$). SnapKV ($n_{lookahead} = 0$) consistently shows the lowest correlation. Experiments use a Qwen2.5-0.5B draft model and a Qwen2.5-32B target model on Multi-news from LongBench.

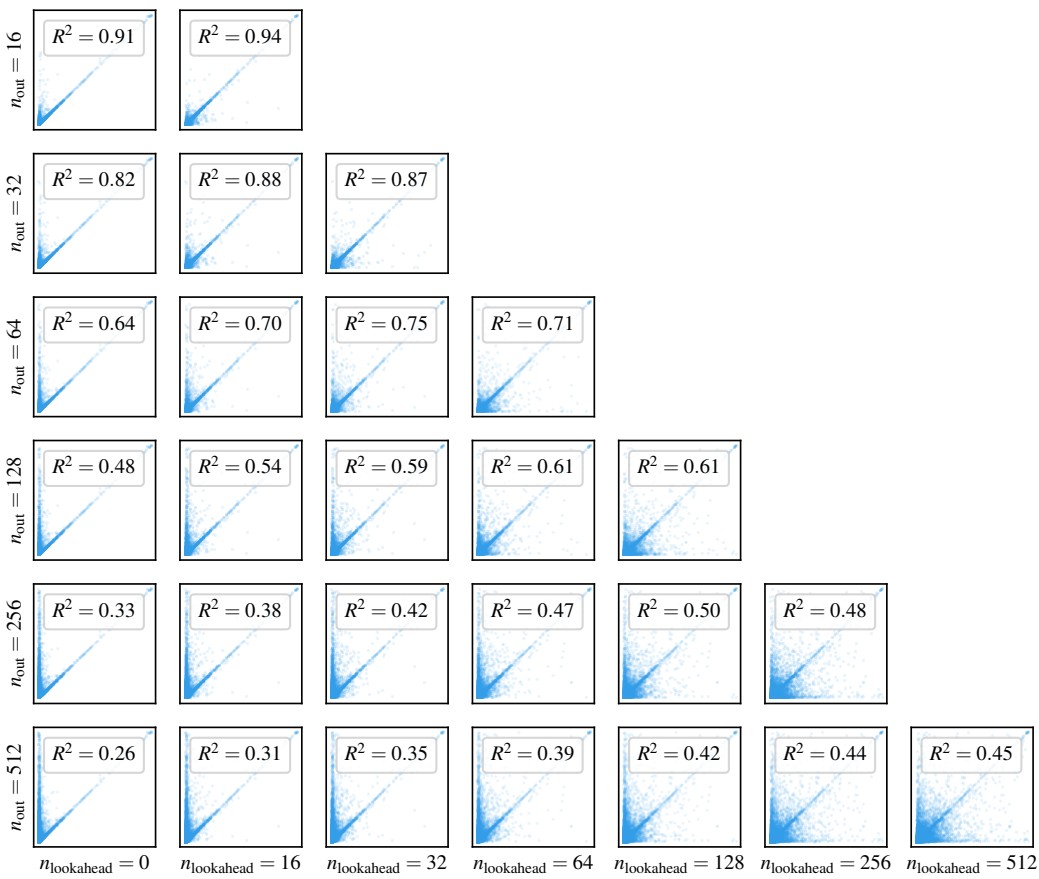

Figure 15: Correlation of importance scores from SpecKV against ground-truth scores from the dense target model for various output lengths ($n_{\text{out}}$) and lookaheads ($n_{\text{lookahead}}$). Increasing $n_{\text{lookahead}}$ improves correlation, especially for longer output lengths ($n_{\text{out}}$). SnapKV ($n_{\text{lookahead}} = 0$) shows the lowest correlation. Experiments use a Qwen2.5-0.5B draft model and a Qwen2.5-32B target model on GovReport from LongBench.

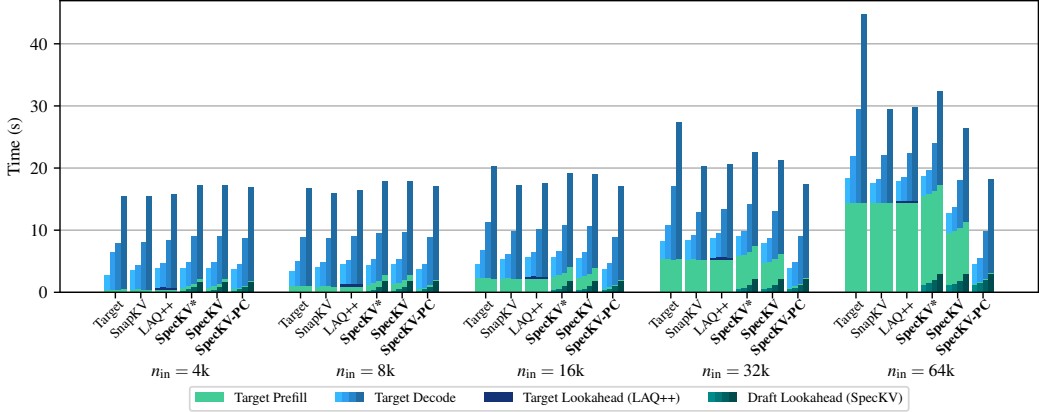

Figure 16: End-to-end latency comparison across five input sequence lengths ($n_{\text{in}}$) and four output sequence lengths ($n_{\text{out}} \in \{64, 128, 256, 512\}$). All experiments use the Qwen2.5 model (3B draft, 32B target) with $C_{\text{max}} = 256$. For SpecKV algorithms, we use $n_{\text{lookahead}} = n_{\text{out}}$. Each bar segment breaks down the latency by processing stage, and the four bars for each method correspond to the four $n_{\text{out}}$ values. SpecKV* denotes SpecKV without sparse prefill, while SpecKV-PC combines SpecKV with SpecPC. Notably, when $n_{\text{in}} \geq 16\text{k}$, the efficiency gains from sparse prefill (SpecKV) and prompt compression (SpecKV-PC) effectively offset the lookahead overhead for $n_{\text{lookahead}} \leq 64$ and $\leq 512$, respectively. Overall, SpecKV-PC is the most efficient method; by precompressing the prompt, it significantly cuts target prefill time, resulting in a speedup of about 40% to 75% (depending on output length) over LAQ++ at a 64k input context.

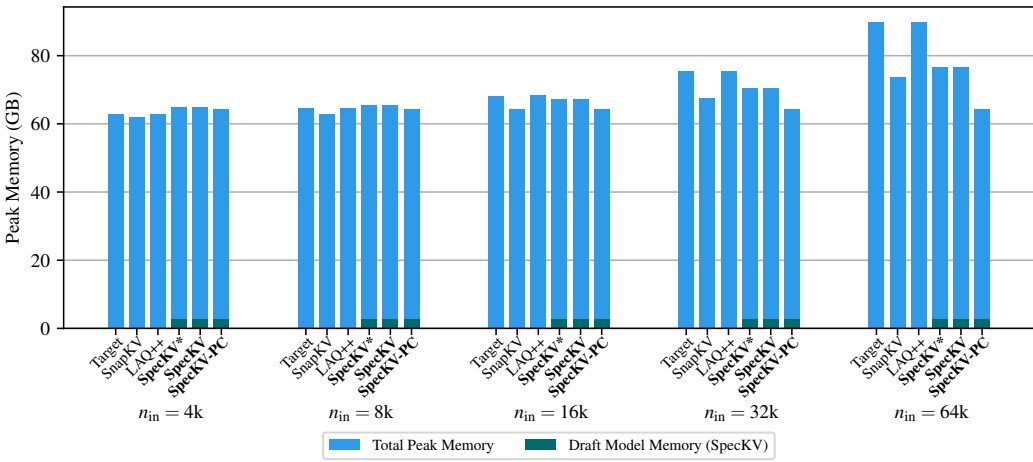

Figure 17: Peak memory usage across five input sequence lengths ($n_{\text{in}}$) using the Qwen2.5 model (3B draft, 32B target) with $C_{\text{max}} = 256$ and $n_{\text{lookahead}} = n_{\text{out}}$. SpecKV* denotes SpecKV without sparse prefill, while SpecKV-PC combines SpecKV with SpecPC. LAQ++ offers no peak memory savings over the target model, as it must store the full cache. SpecKV is slightly more memory-hungry than SnapKV because it also stores the draft model weights (indicated in green), though this constitutes a small fraction of the total memory. SpecKV-PC is the most memory-efficient method; it first compresses the prompt (to 2048 tokens here) before the target model's prefill phase, significantly reducing peak usage. At a 64k context length, SpecKV-PC saves approximately 9 GB over SnapKV and 25 GB over LAQ++.

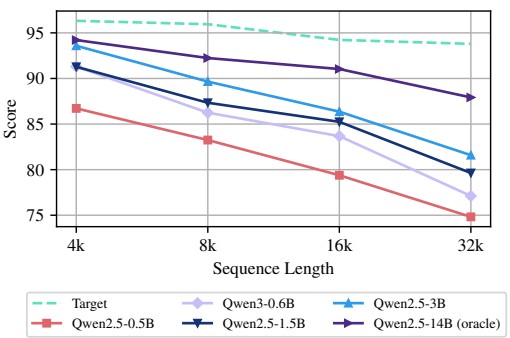

Figure 18: Effect of draft model quality on RULER score. The target model is Qwen2.5-14B (Instruct). Consistent with expectations, employing more capable draft models boosts performance. For reference, we also evaluate an oracle setting where the draft model is identical to the target (Qwen2.5-14B), representing an empirical upper bound for SpecKV.

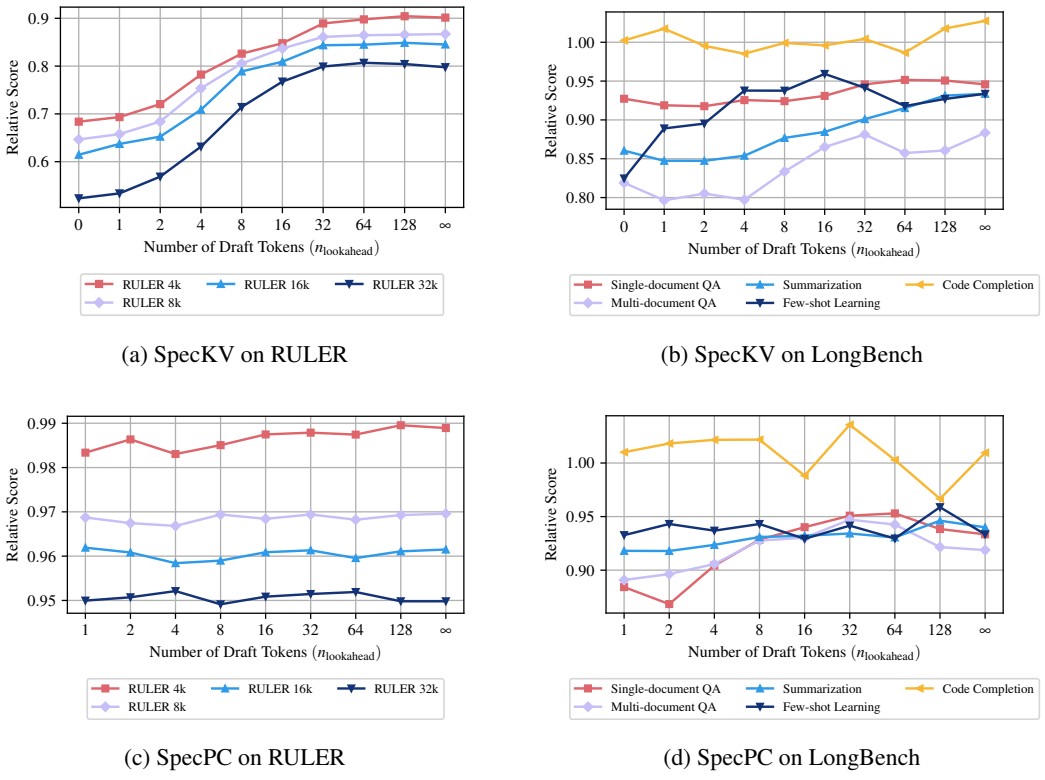

Figure 19: Impact of the number of generated draft tokens, $n_{\text{lookahead}}$, on the relative performance of SpecKV and SpecPC, using Qwen2.5-0.5B-Instruct as the draft model and Qwen2.5-14B-Instruct as the target model. The relative score is calculated as the score of each SpecKV configuration divided by the score of the full dense target model. Increasing $n_{\text{lookahead}}$ substantially boosts SpecKV's score, whereas SpecPC shows only minor improvement with higher $n_{\text{lookahead}}$. SnapKV ($n_{\text{lookahead}} = 0$) has the lowest performance in most cases. $\infty$ denotes lookahead to the EOS token.

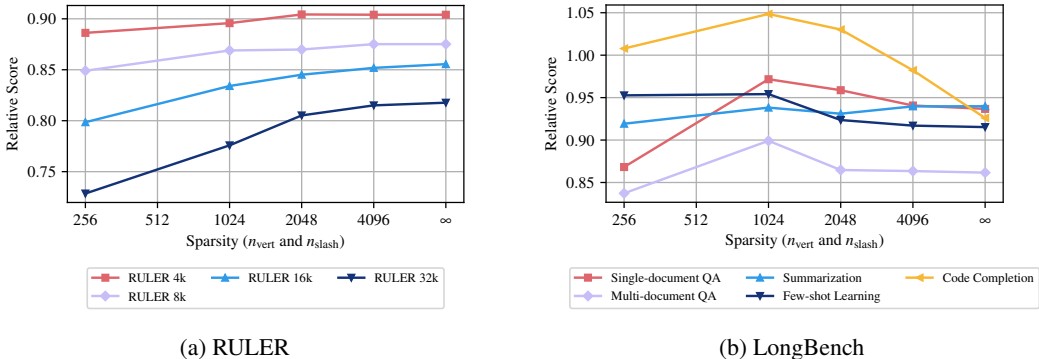

(a) RULER

(b) LongBench

Figure 20: Impact of different sparsity levels in SpecKV's sparse prefill on relative performance, using Qwen2.5-0.5B-Instruct as the draft model and Qwen2.5-14B-Instruct as the target model. We vary $n_{\text{vert}}$ (set equal to $n_{\text{slash}}$) and observe the impact on accuracy. Accuracy improves as sparsity decreases (higher $n_{\text{vert}}$) up to 2048, beyond which gains saturate, hence our choice of 2048 for main results. $\infty$ corresponds to fully dense prefill. Notably, for some LongBench tasks, higher sparsity actually benefits accuracy. The relative score is calculated as the score of each SpecKV configuration divided by the score of the full dense target model.

