# OpenReview forum: "Draft-based Approximate Inference for LLMs"
_ICLR.cc/2026/Conference — ICLR 2026 Poster_

### Official Review · Reviewer_tK1Y · 2025-10-26

**Soundness:** 2
**Presentation:** 3
**Contribution:** 2
**Rating:** 6
**Confidence:** 3

**Summary:**

This paper proposes Draft-based Approximate Inference (DAI), a framework that leverages a small draft model to generate a short sequence of lookahead tokens, which are then used to estimate token or KV importance in large language models (LLMs).
Under this framework, the authors instantiate two methods:

SpecKV — for lookahead-guided KV cache dropping

SpecPC — for prompt compression

The key idea is to use the draft model’s predicted future queries to obtain a more accurate estimate of which tokens will be attended to in future steps, enabling more effective prefill-time cache compression.
Empirical results on RULER and LongBench show that the proposed methods outperform prior works such as SnapKV, LAQ++, and SpecPrefill, with reduced memory and latency.
The paper also provides theoretical bounds relating draft embedding quality to attention approximation error.

**Strengths:**

1.  Clean and implementable idea:
Extends speculative decoding to approximate inference in a conceptually neat way, requiring minimal system changes.

2.  Strong empirical results:
On both synthetic and long-context benchmarks, SpecKV yields clear gains (1–2 points improvement) over existing KV compression baselines, with ~40–50% memory savings.

3.  Solid theoretical justification:
Provides error bounds connecting the draft model’s embedding deviation to importance estimation error, filling a theoretical gap that earlier methods (e.g., SnapKV, LAQ++) lacked.

4.  System efficiency:
The overhead of lookahead generation is minimal (<10% in prefill), while decoding latency and memory are significantly reduced.

**Weaknesses:**

1. Limited conceptual novelty:
The main idea—using a smaller model to predict future attention patterns—is a natural extension of speculative decoding (FastGen, Medusa) and prior KV-dropping methods (SnapKV, LAQ++).
The improvement lies mainly in integration and theoretical refinement, not in introducing a new paradigm.
2. Marginal accuracy gains:
Improvements on benchmarks are moderate (1–2%), suggesting the practical benefit mainly comes from efficiency rather than substantial modeling advance.

**Questions:**

How sensitive are the results to $n_{lookahead}$? The ablation in Appendix E.4 is informative, but further scaling analysis would be valuable.

---

> ### Author Response · Authors · 2025-11-20
>
> We thank the reviewer for the insightful feedback and for recognizing that our work presents a clean, easily implementable approach that achieves consistent empirical improvements across benchmarks. We also appreciate the reviewer's acknowledgment of our theoretical analysis and our substantial reductions in decoding latency and memory usage.
> ***
> >**The main idea—using a smaller model to predict future attention patterns—is a natural extension of speculative decoding and prior KV dropping methods. The improvement lies mainly in integration and theoretical refinement, not in introducing a new paradigm.**
>
> We agree that our main contributions lie in unifying and providing theoretical justification for existing methods. However, we believe that these contributions, together with our algorithmic refinements, make our work valuable.
>
> >**Marginal accuracy gains: Improvements on benchmarks are moderate (1–2%), suggesting the practical benefit mainly comes from efficiency rather than substantial modeling advance.**
>
> While we agree that gains on some benchmarks are in the 1-2% range, we would like to highlight two points that are central to our work:
>
> 1. SpecKV achieves significantly stronger results on RULER, outperforming LAQ++ by up to 10 points ([Figure 5](https://anonymous.4open.science/r/speckv/f05.png)).
> 2. The memory efficiency of SpecKV is a core architectural contribution. LAQ++ uses $O(Ln_\text{in})$ space, since it requires storing the entire KV cache for all $L$ layers between its two forward passes (see the last response to `QfmW` for more details). In contrast, SpecKV uses $O(\max(n_\text{in},LC_\max))$ space, needing to store the full KV cache for just a single layer at a time.
>
> As shown in [Figure 17](https://anonymous.4open.science/r/speckv/f17.png), this difference dramatically reduces peak memory usage, allowing SpecKV to be significantly more efficient than LAQ++ while maintaining superior accuracy.
>
> >**How sensitive are the results to $n_\text{lookahead}$? The ablation in Appendix E.4 is informative, but further scaling analysis would be valuable.**
>
> Thank you for suggesting a deeper scaling analysis. In response, we have expanded our evaluation of $n_\text{lookahead}$ sensitivity in the updated Appendix E.7 (replacing the previous ablation). Our findings are:
>
> 1. Latency: To address the cost of scaling $n_\text{lookahead}$, we provide a complete latency breakdown in [Figure 16](https://anonymous.4open.science/r/speckv/f16.png). This demonstrates that the draft model latency is minor compared to the cost of prefilling the target model.
>    - Crucially, for inputs where $n_\text{in} \ge 16\text{k}$, SpecKV's sparse prefill effectively offsets the lookahead overhead for $n_\text{lookahead} \le 64$.
>    - Similarly, for inputs where $n_\text{in} \ge 16\text{k}$, the combined SpecKV-PC method (utilizing prompt compression) effectively offsets this cost for $n_\text{lookahead} \le 512$.
>    - At 64k input length, SpecKV-PC method remains highly efficient, achieving a 40% speedup and 25 GB memory reduction ([Figure 17](https://anonymous.4open.science/r/speckv/f17.png)) over LAQ++.
> 2. Accuracy: As shown in Appendix E.4 and [Figures 7](https://anonymous.4open.science/r/speckv/f07.png), [14](https://anonymous.4open.science/r/speckv/f14.png), [15](https://anonymous.4open.science/r/speckv/f15.png), and [19](https://anonymous.4open.science/r/speckv/f19.png), increasing $n_\text{lookahead}$ improves correlation and accuracy. Setting $n_\text{lookahead}$ to even a small fraction of $n_\text{out}$ yields improved accuracy over non-lookahead baselines.

---

### Official Review · Reviewer_QfmW · 2025-10-28

**Soundness:** 3
**Presentation:** 3
**Contribution:** 3
**Rating:** 4
**Confidence:** 3

**Summary:**

This paper proposes a framework called draft-based approximate inference to optimize the efficiency of LLM in long context decoding. Existing methods, which approximate KV-cache discarding, sparse attention, and prompt compression, usually rely on rough prediction when estimating the importance of tokens or KV pairs. In contrast, the core idea of this paper is to use a small and cheap "draft model" to generate approximate future output (lookahead), so as to more accurately predict the importance of the current token or KV pair.

In this paper, two methods are introduced:

1. SpecKV, whose goal is to use the "look ahead" capability of the draft model to more accurately discard unimportant parts in the kV cache, and combine sparse pre-filling to improve efficiency.

2. SpecPC, whose goal is to use the draft model to directly determine which tokens in the input prompt are not important, and compress the prompt before sending it to the target model.


A large number of experiments on the benchmark of rule and longbench show that speckv and specpc continuously achieve higher accuracy than the existing baseline methods under the fixed KV cache or prompt size limit.

**Strengths:**

1. This paper provides theoretical (Theorems 1 and 2) and experimental evidence to support the effectiveness of the "look ahead" based importance estimation.

2. This framework unifies and extends the idea of using approximate future information to improve token importance estimation.

3. The paper claims that its method achieves the current state-of-the-art accuracy in the long context benchmark under the constraint of a fixed KV cache or prompt size.

4. Even if a weak draft model is used, this paper can perform well, far exceeding the performance of the draft model itself. Also, using a better draft model will further improve performance.

**Weaknesses:**

1. The core of the whole framework is to use the draft model to approximate the behavior of the target model. Both theoretical analysis (Theorem 1) and experimental results (Fig. 10) show that the accuracy of the draft model directly affects the final performance. If a draft model that is small (low overhead) and similar enough to the target model (high accuracy) cannot be found, the effect of SpecKV and SpecPC may be compromised.

2. Compared with methods such as SnapKV, which only pre-fills and compresses the target model once, the pre-filling steps of SpecKV are more complex, and the calculation cost is higher. Although the paper claims that the overall delay is reduced, this is mainly the benefit of the decoding stage.

3. This paper also introduces a new memory occupation: it needs to load and store the weight of the draft model. Although this overhead is fixed and does not increase with the sequence length as KV cache, it is still an additional memory burden compared with methods that do not require a draft model.

4. The main idea is similar to speculative decoding and previous LAQ++. It is more likely a technical extension.


I have a borderline opinion on this paper and hope to see the rebuttal and other reviewers' comments.

**Questions:**

How does SpecKV reduce the peak memory than LAQ++? In Algorithm 1, the target model still needs to store all the KV cache of the input sequence x and draft output y_draft.

---

> ### Author Response · Authors · 2025-11-20
>
> We thank the reviewer for the thoughtful critique and for recognizing our unified framework and that our methods are supported by clear theoretical and experimental evidence.
> ***
> >**The core of the whole framework is to use the draft model to approximate the behavior of the target model. Both theoretical analysis (Theorem 1) and experimental results (Fig. 10) show that the accuracy of the draft model directly affects the final performance. If a draft model that is small (low overhead) and similar enough to the target model (high accuracy) cannot be found, the effect of SpecKV and SpecPC may be compromised.**
>
> We agree with the reviewer's assessment. The draft model's accuracy is indeed crucial for performance. We consider this requirement to be highly practical for two key reasons. First, the rise of models for speculative decoding provides a rich set of off-the-shelf draft candidates. Furthermore, as demonstrated in the new Appendix Section E.5 ([Figure 8](https://anonymous.4open.science/r/speckv/f08.png), [Table 7](https://anonymous.4open.science/r/speckv/t07.png)), our approach supports draft models from different model families. Second, model distillation can be an effective way to train draft models that closely match the behavior of the target model, for cases where an ideal off-the-shelf model is not available. We have updated the Limitations section to explicitly acknowledge this requirement.
>
> >**Compared with methods such as SnapKV, which only pre-fills and compresses the target model once, the pre-filling steps of SpecKV are more complex, and the calculation cost is higher. Although the paper claims that the overall delay is reduced, this is mainly the benefit of the decoding stage.**
>
> We thank the reviewer for highlighting this trade-off. We acknowledge that the drafting steps in SpecKV introduce additional computational overhead compared to purely compression-based methods like SnapKV. However, our empirical results show that this overhead is negligible or effectively offset in long context settings.
>
> To demonstrate this, we have added a full latency breakdown in Section E.7 ([Figure 16](https://anonymous.4open.science/r/speckv/f16.png)). Additionally, we introduce SpecKV-PC, which combines SpecKV with SpecPC (prompt compression) to further minimize latency.
>
> Our new analysis (setting $n_\text{lookahead}=n_\text{out}$ for SpecKV) reveals three key findings:
>
> - Crucially, for inputs where $n_\text{in} \ge 16\text{k}$, SpecKV's sparse prefill effectively offsets the lookahead overhead for $n_\text{lookahead} \le 64$.
> - Similarly, for inputs where $n_\text{in} \ge 16\text{k}$, the combined SpecKV-PC method (utilizing prompt compression) effectively offsets this cost for $n_\text{lookahead} \le 512$.
> - At 64k input length, SpecKV-PC method remains highly efficient, achieving a 40% speedup and 25 GB memory reduction ([Figure 17](https://anonymous.4open.science/r/speckv/f17.png)) over LAQ++.
>
> These empirical results confirm that SpecKV's prefill, even with the drafting component, is as efficient, or more efficient, than SnapKV in practice.
>
> >**Although the memory overhead of draft model weights is fixed and does not increase with the sequence length, it is still an additional memory burden compared with methods that do not require a draft model.**
>
> While draft models add memory overhead, this can be mitigated. In single-instance settings, draft weights can be offloaded to CPU memory during the target model's forward pass. In multi-instance settings, memory is saved by serving fewer draft models than target models, since their high speed allows them to keep pace. Finally, the model choice itself involves a quality-vs-memory trade-off, which can be further tuned using weight quantization.
>
> [Figure 17](https://anonymous.4open.science/r/speckv/f17.png) confirms that the draft weights constitute only a minor portion of the total peak memory. The figure also highlights that the combined SpecPC-KV method significantly reduces peak memory. This reduction is achieved by compressing the prompt first so that only a portion is fed to the target model, resulting in a 25 GB memory saving compared to LAQ++ at 64k context.

---

> ### Author Response · Authors · 2025-11-20
>
> >**The main idea is similar to speculative decoding and previous LAQ++. It is more likely a technical extension.**
>
> We agree that our work draws inspiration from speculative decoding and previous LAQ++ methods, but it introduces key differences that make it distinct and valuable.
>
> While both our methods and speculative decoding use small draft models to accelerate inference, their goals differ. Speculative decoding increases utilization without altering the output distribution but does not reduce overall memory or computation complexity. In contrast, SpecKV and SpecPC achieve greater latency and memory efficiency by approximating outputs rather than reproducing them exactly. Additionally, speculative decoding can be used on top of SpecKV or SpecPC to further increase generation throughput.
>
> Although SpecKV’s main idea is related to LAQ++, it directly addresses the major limitation of LAQ++ described below.
>
> >**How does SpecKV reduce the peak memory than LAQ++? In Algorithm 1, the target model still needs to store all the KV cache.**
>
> While SpecKV and LAQ++ use the same amount of memory when examining a single attention layer, their memory usage across the entire prefill stage is different.
>
> SpecKV compresses the KV cache within a single forward pass, processing the model one layer at a time. It only needs to temporarily store the full KV cache for a single layer (using $O(n_\text{in})$ space) before compressing it. The final, persistent storage for the compressed cache across all $L$ layers requires $O(LC_\max)$ space. Because SpecKV never holds the full cache for all layers simultaneously, its peak prefill space complexity is $O(\max(n_\text{in}, LC_\max))$.
>
> In contrast, LAQ++ uses a two-pass approach. It first generates draft queries using an initially compressed cache. These queries are then used to compute more accurate importance scores for a second, more informed compression pass. Because the final compression does not happen in a single pass, LAQ++ must store the entire, uncompressed KV cache for all layers between the two passes, requiring $O(Ln_\text{in})$ space. While this high memory cost could be mitigated by recomputing the KV cache from scratch during the second pass, this alternative would effectively double the prefill time.

---

> > ### Comment · Reviewer_QfmW · 2025-11-24
> >
> > Thank you. This rebuttal helps a lot. I am still concerned that this paper is a technical extension. I will take some time to check other reviewers' comments for a final decision.

---

> ### Author Response · Authors · 2025-11-25
> **Clarification on Novelty: Theoretical Foundations & Framework Generality**
>
> We thank the reviewer for the continued engagement. As you finalize your decision, we wish to briefly address the "technical extension" concern by highlighting three fundamental shifts our framework introduces over prior work (like LAQ++):
>
> **1. Theoretical Foundation (Filling the Gap):**
> Prior works operate as heuristics, demonstrating *that* lookahead works, but not *why*. As noted by Reviewer `tK1Y`, our work fills this gap. Theorems 1 & 2 provide the first rigorous bounds linking draft quality to approximation error, elevating the field from empirical heuristics to **principled approximation with provable error bounds**.
>
> **2. Solving the Structural Bottleneck ($O(Ln_\text{in})$ vs $O(\max(n_\text{in}, LC_\max))$):**
> LAQ++ is structurally limited by an $O(L n_\text{in})$ prefill memory bottleneck. SpecKV structurally resolves this by employing a separate small draft model. The lightweight draft cache is released immediately after lookahead generation, enabling the second pass to compress the target model KV cache layer-by-layer ($O(\max(n_\text{in}, LC_\max))$). This makes lookahead-based KV compression viable for real-world scenarios.
>
> **3. (New) Framework Generality (SpecKV-PC):**
> Our framework enables **cascaded compression**. By composing SpecPC and SpecKV (SpecKV-PC method), we achieve efficiency gains superior to baselines like LAQ++ (e.g., **40% speedup and 25GB memory reduction** at 64k context), while often exceeding the accuracy of using SpecKV alone.
>
> We believe these contributions, establishing the theoretical laws, solving the memory bottleneck, and enabling cascading compression, distinguish this work as a foundational framework rather than an incremental extension.

---

> > ### Comment · Reviewer_QfmW · 2025-11-25
> >
> > Thank you! These further rebuttals do persuade me. I decide to raise my score.

---

> ### Author Response · Authors · 2025-11-25
> **Thank You**
>
> We thank the reviewer for their time, engagement with our rebuttal, and reconsideration of the score. We appreciate the constructive feedback throughout this process, which has helped strengthen the paper.

---

### Official Review · Reviewer_bFv3 · 2025-10-31

**Soundness:** 3
**Presentation:** 3
**Contribution:** 3
**Rating:** 8
**Confidence:** 3

**Summary:**

This paper proposes an effective "Draft-based Approximate Inference" framework that leverages a draft model for lookahead to more accurately estimate the importance of tokens and KV pairs, thereby improving long-context approximate inference. The authors further introduce SpecKV and SpecPC for KV cache dropping and prompt compression, respectively, and provide a thorough error analysis. A comprehensive set of experiments further demonstrates the effectiveness and application potential of the proposed method.

**Strengths:**

1. By using a draft model to estimate the importance of tokens in the KV cache and prompt, the method achieves strong performance under controllable complexity.
2. This work provide a clear theoretical analysis, demonstrating how embedding errors influence KV importance estimation errors (Theorem 1), and how output approximation under RIP or more general assumptions can upper bound attention approximation errors (Theorems 2 and 3).
3. The experiments on LongBench and RULER Benchmarks are solid, with results that convincingly demonstrate the effectiveness of the proposed methods.

**Weaknesses:**

1. For different input embeddings, are there any limitations to the applicability of Theorem 2?
2. There appear to be some typo errors in Table 2.

**Questions:**

1. I am curious whether using different types of draft models and target models would affect token selection (e.g., Qwen-2.5-0.5B + Llama-3-70B). Does this imply that draft models must be selected from the same model family as the target model?
2. As shown in Figure 2, the benefits gained from the draft model vary across different tasks. Any explaination?

---

> ### Author Response · Authors · 2025-11-20
>
> We thank the reviewers for the thoughtful feedback and for recognizing that our methods are supported by clear theoretical analysis, and show strong empirical performance.
> ***
> >**For different input embeddings, are there any limitations to the applicability of Theorem 2?**
>
> Yes, with specific input embeddings, Theorem 2 does not hold. A specific failure case is the first layer of a transformer when there are duplicate input embeddings. However, this limitation is less significant for deeper transformer layers.
>
> >**There appear to be some typo errors in Table 2.**
>
> We thank the reviewer for the careful reading. We updated names to be consistent throughout the paper.
>
> >**I am curious whether using different types of draft models and target models would affect token selection (e.g., Qwen-2.5-0.5B + Llama-3-70B). Does this imply that draft models must be selected from the same model family as the target model?**
>
> From a technical compatibility perspective, neither SpecKV nor SpecPC imposes constraints on the choice of draft and target models, as the interaction between the two occurs through a text-only interface. In SpecKV, the draft output text is used to improve KV-pair importance estimation in the target model. In SpecPC, the draft model compresses the input text, and the resulting compressed prompt is passed to the target model. To demonstrate this, we evaluate SpecKV and SpecPC using draft and target models from different model families in Appendix Section E.5. We find that while both algorithms function across model families ([Figure 8](https://anonymous.4open.science/r/speckv/f08.png) and [Table 7](https://anonymous.4open.science/r/speckv/t07.png)), SpecKV performs better because it has fewer requirements regarding model similarity. Specifically, SpecKV only requires that the draft and target outputs be similar, whereas SpecPC requires that the draft and target attention scores be similar.
>
> >**As shown in Figure 2, the benefits gained from the draft model vary across different tasks. Any explanation?**
>
> We assume the reviewer refers to Table 2: LongBench performance with Qwen2.5 and Llama-3. We believe the variation in benefits gained from the draft model across different tasks may be explained by two factors. First, some tasks may not require lookahead to identify which parts of the context are important. In such cases, adding draft model lookahead is unlikely to improve accuracy. Second, if the draft model is too weak for a given task or its outputs differ significantly from those of the target model, its outputs will not be useful for lookahead.
>
> Additionally, we direct the reviewer to our updated [Figure 19](https://anonymous.4open.science/r/speckv/f19.png), which presents an extended ablation study on the impact of $n_\text{lookahead}$. By showing accuracy relative to the target model, we make the following observation: SpecKV gains less benefit from additional lookahead tokens on tasks where the SnapKV baseline ($n_\text{lookahead} = 0$) already performs well (relative score $\approx 1$).

---

### Official Review · Reviewer_zgDL · 2025-10-31

**Soundness:** 2
**Presentation:** 3
**Contribution:** 3
**Rating:** 4
**Confidence:** 3

**Summary:**

This paper proposes Draft-based Approximate Inference, leveraging lightweight draft models for lookahead-based token/KV importance estimation. It introduces SpecKV for KV cache dropping with sparse prefill and SpecPC for prompt compression, both supported by theoretical proofs and extensive experiments showing superior accuracy–efficiency trade-offs over baselines.

**Strengths:**

1. First integration of draft-model lookahead into KV dropping and prompt compression, with theoretical justification.
2. Strong empirical gains across diverse benchmarks, models, and compression budgets.
3. Clear motivation, concise algorithms, and well-presented results.

**Weaknesses:**

1. Lacks analysis of importance score differences with/without lookahead
2. Limited breakdown of the latency trade-off
3. Unclear whether SpecKV and SpecPC can be effectively combined.

**Questions:**

1. The core premise is that approximate future information improves token/KV importance estimation. Could the authors present a quantitative comparison of importance score distributions obtained with lookahead versus current-token-only methods? If the distributions differ substantially, would SpecKV’s advantage over methods like SnapKV diminish when output length greatly exceeds $n_{lookahead}$?
2. The acceleration analysis is limited. Since SpecKV and SpecPC incur additional draft-model overhead, can the authors provide a detailed latency breakdown (draft inference, dense/sparse prefill, decoding) for varying input/output lengths?
3. Given that SpecKV employs sparse prefill, has its speed been compared directly to optimized prefill approaches such as MInference [1]?
4. Can SpecKV and SpecPC be combined in a single pipeline, and if so, could the authors include ablation studies showing the individual and combined contributions to speed, memory reduction, and accuracy?

[1] MInference 1.0: Accelerating Pre-filling for Long-Context LLMs via Dynamic Sparse Attention

---

> ### Author Response · Authors · 2025-11-20
>
> We thank the reviewer for the insightful feedback and for recognizing that our work is the first to integrate draft-model lookahead with KV dropping, offering theoretical grounding, robust empirical gains, and well-defined motivation, algorithms, and results. We greatly appreciate the comment regarding the combination of SpecKV and SpecPC.
> ***
> >**Could the authors present a quantitative comparison of importance scores obtained with lookahead versus current-token-only methods? How do they compare when the output length greatly exceeds $n_\text{lookahead}$?**
>
> The new Section E.4, along with [Figures 14](https://anonymous.4open.science/r/speckv/f14.png) and [15](https://anonymous.4open.science/r/speckv/f15.png), compares SpecKV importance scores against ground-truth scores from the target model. This comparison evaluates different $n_\text{lookahead}$ and $n_\text{out}$ values, including current-token-only methods ($n_\text{lookahead} = 0$). Results show that lookahead yields better correlation, particularly for longer output lengths $n_\text{out}$. This is further supported by [Figure 7](https://anonymous.4open.science/r/speckv/f07.png), which shows that increasing lookahead improves $R^2$ and downstream scores. Notably, SpecKV enhances token importance estimation and accuracy even when $n_\text{lookahead}$ is just 6.25% of $n_\text{out}$. Finally, the updated [Figure 19](https://anonymous.4open.science/r/speckv/f19.png) confirms that lookahead leads to better downstream accuracy on LongBench and RULER.
>
> >**The acceleration analysis is limited. Since SpecKV and SpecPC incur additional draft-model overhead, can the authors provide a detailed latency breakdown (draft inference, dense/sparse prefill, decoding) for varying input/output lengths?**
>
> We thank the reviewer for this suggestion. To address this, we add Section E.7 and [Figure 16](https://anonymous.4open.science/r/speckv/f16.png), which provide a detailed latency breakdown across varying algorithms, input lengths ($n_\text{in}$), and output lengths.These new results confirm that the draft model latency remains minor relative to the target model's prefill cost. For inputs where $n_\text{in} \ge 16\text{k}$, sparse prefill (SpecKV) and prompt compression (SpecKV-PC) effectively offset lookahead overheads for $n_\text{lookahead} \le 64$ and $\le 512$, respectively. Consequently, SpecKV-PC remains highly efficient, achieving a **40% speedup** and **25 GB memory reduction** over LAQ++ at 64k context ([Figure 17](https://anonymous.4open.science/r/speckv/f17.png)).
>
> >**Given that SpecKV employs sparse prefill, has its speed been compared directly to optimized prefill approaches such as MInference?**
>
> Since SpecKV directly employs the Vertical-Slash sparse attention kernel from MInference [1], its speed matches MInference when the same levels of sparsity are applied. The only performance difference stems from overhead introduced by the draft model. As shown in [Figure 16](https://anonymous.4open.science/r/speckv/f16.png), this overhead is minor, especially when $n_\text{lookahead}$ is small.
>
> >**Can SpecKV and SpecPC be combined in a single pipeline, and if so, could the authors include ablation studies showing the individual and combined contributions to speed, memory reduction, and accuracy?**
>
> We thank the reviewer for this excellent suggestion. This is a key point, and the answer is yes, they can be combined, and we found this approach to be highly effective.
>
> In response, we add SpecKV-PC to the revised paper (Sections 4.3, 5, and E.6). Our findings regarding the combined contributions are as follows:
>
> 1. Pipeline Design: We implement a sequential approach where we first compress the prompt with SpecPC, and then further compress the KV cache with SpecKV. This adds no additional draft overhead compared to SpecKV alone (Section 4.3, [Figure 4](https://anonymous.4open.science/r/speckv/f04.png)).
> 2. Accuracy: As shown in [Figure 9](https://anonymous.4open.science/r/speckv/f09.png) and [Table 8](https://anonymous.4open.science/r/speckv/t08.png), by selecting the appropriate ratios of SpecPC and SpecKV compression, we can even achieve higher accuracy than with SpecKV alone, particularly at longer context lengths. This indicates that the initial SpecPC-based prompt compression acts as a pre-filter, removing easy-to-identify, unimportant tokens. This improvement is consistent across multiple model families and benchmarks. Given its strong performance, we further update [Figure 5](https://anonymous.4open.science/r/speckv/f05.png) and [Table 2](https://anonymous.4open.science/r/speckv/t02.png) in the main body
> 3. Latency and Peak Memory: The combined approach is substantially more efficient than SpecKV alone. By reducing the initial prompt size, we significantly lower target prefill computation time and overall memory footprint (Section E.7, [Figures 16](https://anonymous.4open.science/r/speckv/f16.png) and [17](https://anonymous.4open.science/r/speckv/f17.png)).

---

### Author Response · Authors · 2025-11-20
**To AC and Reviewers**

We thank the reviewers for the careful and insightful feedback. We are encouraged that they commended our work as the first to integrate draft-model lookahead with KV dropping, highlighting **its novelty and clear motivation** (`zgDL`); its **strong theoretical grounding** (`zgDL`, `bFv3`, `QfmW`, `tK1Y`) and **comprehensive empirical validation** (`zgDL`, `bFv3`, `QfmW`), which yields **strong empirical results** (`zgDL`, `bFv3`, `QfmW`, `tK1Y`); and its value as a **clean, well-motivated, and implementable framework** that consistently improves efficiency and memory usage (`tK1Y`). In response, we made the following key additions to our paper.

## **Major Update 1: Cascaded Compression with SpecKV and SpecPC (SpecKV-PC)**

We introduce a combined **SpecKV-PC** method in Section 4.3 ([Figure 4](https://anonymous.4open.science/r/speckv/f04.png)), which first compresses the prompt with SpecPC and then further compresses the KV cache with SpecKV. For instance, SpecKV-PC-2048 compresses the prompt to 2048 tokens and then the KV-cache to $C_\text{max}=256$.

This integrated approach is highly efficient, leveraging a single draft lookahead to provide both the importance scores for SpecPC and the output tokens for SpecKV. This has two primary benefits. First, it provides substantial latency and memory gains, as the target model only processes a small portion of the original prompt. Compared to LAQ++ at 64k context, SpecKV-PC is **~40% faster** ([Figure 16](https://anonymous.4open.science/r/speckv/f16.png)) and requires **25GB less memory** ([Figure 17](https://anonymous.4open.science/r/speckv/f17.png)). Second, it surprisingly **surpasses the accuracy of SpecKV alone** (Section E.6, [Figure 9](https://anonymous.4open.science/r/speckv/f09.png), [Table 8](https://anonymous.4open.science/r/speckv/t08.png)). This suggests that the initial SpecPC-based prompt compression acts as a pre-filter, removing easy-to-identify, unimportant tokens. Due to its strong performance, we include SpecKV-PC experiments in the main paper ([Figures 5](https://anonymous.4open.science/r/speckv/f05.png) and [6](https://anonymous.4open.science/r/speckv/f06.png), [Table 2](https://anonymous.4open.science/r/speckv/t02.png))

## **Major Update 2: Extended Latency and Memory Analysis**

We detail latency and peak memory usage for each algorithm across various input and output lengths in Appendix E.7.

[Figure 16](https://anonymous.4open.science/r/speckv/f16.png) presents a breakdown of end-to-end latency, dividing it into three stages: draft generation, target prefill, and target decoding. This shows draft latency is minor compared to the cost of prefilling the target model.
   - Crucially, for inputs where $n_\text{in} \ge 16\text{k}$, SpecKV's sparse prefill effectively offsets the lookahead overhead for $n_\text{lookahead} \le 64$.
   - Similarly, for inputs where $n_\text{in} \ge 16\text{k}$, the combined SpecKV-PC method (utilizing prompt compression) effectively offsets this cost for $n_\text{lookahead} \le 512$.
   - At 64k input length, SpecKV-PC method remains highly efficient, achieving a 40% speedup over LAQ++.

[Figure 17](https://anonymous.4open.science/r/speckv/f17.png) shows the peak memory consumption for each algorithm. While our methods need to store draft weights, this cost is minor compared to the total peak memory. Additionally, SpecKV-PC significantly reduces memory consumption, as only a small portion of the prompt is fed to the target model.

## **Major Update 3: Extended Study on the Impact of $n_\text{lookahead}$ in SpecKV**

We add Section E.4, detailing the impact of $n_\text{lookahead}$ on importance score correlation. [Figure 7](https://anonymous.4open.science/r/speckv/f07.png) shows that increasing $n_\text{lookahead}$ in SpecKV improves both its correlation with ground-truth scores and the resulting downstream accuracy, even when $n_\text{lookahead}$ is only 6.25% of $n_\text{out}$. [Figures 14](https://anonymous.4open.science/r/speckv/f14.png) and [15](https://anonymous.4open.science/r/speckv/f15.png) further detail this relationship, plotting the correlation for various output lengths ($n_\text{out}$) and $n_\text{lookahead}$ values. Finally, [Figure 19](https://anonymous.4open.science/r/speckv/f19.png) extends our previous ablation study on $n_\text{lookahead}$, testing the performance of SpecKV and SpecPC with values of $n_\text{lookahead}$ ranging from 0 (equivalent to SnapKV) to 128.

## **Major Update 4: Cross-Family Model Results**

Section E.5 tests the cross-family compatibility of SpecKV and SpecPC, where draft and target models originate from different families. In [Figure 8](https://anonymous.4open.science/r/speckv/f08.png) and [Table 7](https://anonymous.4open.science/r/speckv/t07.png), we test a Llama-3.1-70B target model with different Qwen2.5 variants as draft models. The results highlight that SpecKV, in particular, maintains strong performance in these scenarios and outperforms LAQ++.

---

### Author Response · Authors · 2025-12-02
**Summary for AC: Score Raise (4 → 6) & Major Revisions**

Dear Area Chair,

We understand you have been assigned this paper under exceptional circumstances. As the system freeze prevented reviewers from updating their scores or responding to our revisions, we provide this summary to assist your decision-making process, specifically highlighting (1) a **confirmed score increase** and (2) a **major algorithmic addition (SpecKV-PC)** that directly addresses reviewer feedback.
***
## **Summary of Reviewer Status**
*Current Official Avg: 5.5 → Revised Avg: 6.0*
| Reviewer | Score | Replied before freeze? | Key Outcome |
| :--- | :--- | :--- | :--- |
| **`bFv3`** | **8** | No | Rated "Good Paper"; commended theoretical justification. |
| **`QfmW`** | **4 → 6** | **Yes** | **Explicitly raised score** on Nov 25. |
| **`tK1Y`** | **6** | No | Acknowledged method "fills a theoretical gap." |
| **`zgDL`** | 4 | No | Main request (Combined Pipeline) **implemented**. |
***
## **1. Reviewer `QfmW`: Score Raise (4 → 6)**
* **Initial Concern**: The reviewer initially questioned the framework's novelty and memory constraints.
* **Resolution**: We clarified that our work provides the first theoretical error bounds for this domain (Theorems 1 & 2) and detailed how ours solves the memory bottleneck of prior work.
* **Outcome**: The reviewer explicitly stated:
   > **"Thank you! These further rebuttals do persuade me. I decide to raise my score."** (Nov 25).

***
## **2. Reviewer `zgDL` (Score: 4): Addressed Concerns**
* **Initial Concern**: Requested a "combined pipeline" of our two methods.
* **Resolution (Major Update):** In response, we implemented **SpecKV-PC**, a cascaded compression method (fully added to the revised paper in Section 4.3), achieving a **40% speedup and 25GB memory reduction** over the LAQ++ baseline, significantly strengthening the paper.
* **Other Concerns**: We provided a full breakdown (draft vs. target cost) in [Figure 16](https://anonymous.4open.science/r/speckv/f16.png) and added quantitative comparisons of importance scores with and without lookahead in Appendix E.4.
* **Status**: Reviewer `zgDL` did not reply before the freeze. As we have fully implemented their requests, we believe the grounds for the initial score of 4 are now resolved.
***
## **3. Reviewer `bFv3`: Support (Score: 8)**
* **Context**: Rated the paper "Good Paper," commending the **"solid theoretical justification"** and **"comprehensive experiments"**.
* **Status**: All minor clarification questions were answered.
***
## **4. Reviewer `tK1Y`: Support (Score: 6)**

* **Context**: Gave a positive score. While noting "limited conceptual novelty," they acknowledged our work **"fills a theoretical gap"** previous methods lacked.
* **Resolution**:
   * **On Novelty**: We note that Reviewer `QfmW` initially shared the same novelty concern. However, after we clarified our theoretical framework (Theorems 1 & 2), `QfmW` **explicitly withdrew the concern** and raised their score.
   * **On Accuracy**: We clarified the significant accuracy gains on RULER (>10 points over baselines).
   * **On Scaling**: We added the requested scaling analysis in Appendix E.7.


***
## **Conclusion**
With the **explicit support of `bFv3` (8)**, the **score raise of `QfmW` (4 → 6)**, and the major development of **SpecKV-PC** to address `zgDL`'s feedback, we believe the submission is now significantly stronger than the initial version. We respectfully ask the AC to evaluate the paper based on these concrete improvements.

Sincerely,
The Authors

---

### Meta-Review · Area_Chair_n3FH · 2026-01-13

**Summary:**

Reviewers broadly agreed that the paper presents a technically sound and well-executed framework for long-context approximate inference using draft-model lookahead, supported by solid theory and extensive experiments. The main points of contention centered on:

- Conceptual novelty: Multiple reviewers (QfmW, tK1Y) initially viewed the work as a technical extension of speculative decoding and prior KV-dropping methods (e.g., SnapKV, LAQ++), rather than a fundamentally new paradigm.

- System-level trade-offs: Concerns were raised about draft-model overhead, prefill complexity, latency breakdowns, and peak memory usage relative to baselines (QfmW, zgDL).

- Completeness of the framework: One reviewer (zgDL) explicitly questioned whether SpecKV and SpecPC could be combined into a single, practical pipeline and requested empirical validation.

- Sensitivity and robustness: Reviewers asked for deeper analysis of lookahead length, importance-score quality with/without lookahead, and cross-model-family robustness.

The rebuttal and revised manuscript substantially strengthened the paper by adding a cascaded SpecKV-PC pipeline, expanded latency/memory analysis, new ablations on lookahead, and cross-family experiments. These additions directly addressed the most substantive reviewer concerns, shifting the balance toward a positive decision.

**Reviewer Concerns:**

Concerns Largely Addressed

Combination of methods: The request to combine SpecKV and SpecPC was fully addressed via the new SpecKV-PC method, with clear algorithmic description and strong empirical gains in accuracy, latency, and memory.

Latency and memory breakdown: Detailed end-to-end latency and peak-memory analyses were added, clarifying that draft overhead is minor and that SpecKV structurally reduces peak prefill memory compared to LAQ++.

- Importance of lookahead: Quantitative comparisons of importance scores with and without lookahead, plus extended ablations on lookahead length, directly answered zgDL’s and tK1Y’s questions.

- Cross-family robustness: New experiments demonstrated that draft and target models need not belong to the same family, partially mitigating concerns about draft-model availability.

- Novelty clarification: While the core idea builds on prior work, the rebuttal convincingly argued that the theoretical framework, memory-efficient structure, and cascaded compression elevate the contribution beyond a minor extension. This persuaded at least one initially skeptical reviewer (QfmW).

Concerns Partially or Still Outstanding

- Fundamental novelty: Some reviewers (notably tK1Y) may still view the work as more incremental than paradigm-shifting, despite the stronger framing and theory.

- Dependence on draft-model quality: Although acknowledged and discussed as a limitation, the reliance on sufficiently aligned draft models remains an inherent assumption rather than a fully resolved issue.

Overall, remaining concerns are largely judgment-based rather than technical, and do not point to clear flaws in correctness or evaluation.

**Reviewer Scores:**

Expected Reviewer Score Changes After Full Discussion

Reviewer bFv3 (initial 8): Likely unchanged at 8. This reviewer was already strongly positive and had only minor clarification questions.

Reviewer QfmW (initial 4 → explicitly raised to 6): Confirmed increase to 6, following rebuttal and novelty clarification.

Reviewer tK1Y (initial 6): Likely to remain at 6, possibly leaning slightly more positive due to added scaling and latency analyses, but still tempered by novelty concerns.

Reviewer zgDL (initial 4): Likely to increase to 6 (or at least borderline accept), as all explicitly requested items—importance-score analysis, latency breakdown, and a combined pipeline—were fully implemented.

---

### Decision · Program_Chairs · 2026-01-26

Accept (Poster)